# Posterior Behavioral Cloning: Pretraining BC Policies for Efficient RL Finetuning

Andrew Wagenmaker [1]   Perry Dong [2]   Raymond Tsao [1]   Chelsea Finn [2]   Sergey Levine [1]

## Abstract

Standard practice across domains from robotics to language is to first pretrain a policy on a large-scale demonstration dataset, and then finetune this policy, typically with reinforcement learning (RL), in order to improve performance on deployment domains. This finetuning step has proved critical in achieving human or super-human performance, yet while much attention has been given to developing more effective finetuning algorithms, little attention has been given to ensuring the pretrained policy is an effective initialization for RL finetuning. In this work we seek to understand how the pretrained policy affects finetuning performance, and how to pretrain policies in order to ensure they are effective initializations for finetuning. We first show theoretically that standard behavioral cloning (BC) can fail to ensure coverage over the demonstrator's actions, a minimal condition necessary for effective RL finetuning. We then show that if, instead of exactly fitting the observed demonstrations, we train a policy to model the *posterior* distribution of the demonstrator's behavior given the demonstration dataset, we *do* obtain a policy that ensures coverage over the demonstrator's actions, enabling more effective finetuning. Furthermore, this policy achieves this while ensuring pretrained performance is no worse than that of the BC policy. We then show this approach is practically implementable with modern generative models and leads to significantly improved RL finetuning performance on both realistic robotic control benchmarks and real-world robotic manipulation tasks, as compared to standard behavioral cloning.

[1]Department of Electrical Engineering & Computer Sciences, UC Berkeley, Berkeley, CA, USA [2]Department of Computer Science, Stanford University, Stanford, CA, USA. Correspondence to: Andrew Wagenmaker <ajwagen@berkeley.edu>.

*Proceedings of the 43rd International Conference on Machine Learning*, Seoul, South Korea. PMLR 306, 2026. Copyright 2026 by the author(s).

## 1. Introduction

Across domains—from language, to vision, to robotics—a common paradigm has emerged for training highly effective "policies": collect a large set of demonstrations, "pretrain" a policy via behavioral cloning (BC) to mimic these demonstrations, then "finetune" the pretrained policy on a deployment domain of interest. While pretraining can endow the policy with generally useful abilities, the finetuning step has proved critical in obtaining effective performance, enabling human value alignment and reasoning capabilities in language domains (Ouyang et al., 2022; Bai et al., 2022a; Team et al., 2025; Guo et al., 2025a), and improving task solving precision and generalization to unseen tasks in robotic domains (Nakamoto et al., 2024; Chen et al., 2025c; Kim et al., 2025; Wagenmaker et al., 2025a). In particular, reinforcement learning (RL)-based finetuning—where the pretrained policy is deployed in a setting of interest and its behavior updated based on the outcomes of these online rollouts—is especially crucial in improving the performance of a pretrained policy.

Critical to achieving successful RL-based finetuning performance in many domains—particularly in settings when policy deployment is costly and time-consuming, such as robotic control—is sample efficiency: effectively modifying the behavior of the pretrained model using as few deployment rollouts as possible. While significant attention has been given to developing more efficient finetuning algorithms, this ignores a primary ingredient in the RL finetuning process: the pretrained policy itself. Though generally more effective pretrained policies are the preferred initialization for finetuning (Guo et al., 2025a; Yue et al., 2025), it is not well understood how pretraining impacts finetuning performance beyond this, and how we might pretrain policies to enable more efficient RL finetuning.

In this work we seek to understand the role of the pretrained policy in RL finetuning, and how we might pretrain policies that (a) enable efficient RL finetuning, and (b) before finetuning, perform no worse than the policy pretrained with standard BC. We propose a novel pretraining approach—*posterior behavioral cloning* (POSTBC)—which, rather than fitting the empirical distribution of demonstrations as standard BC does, instead fits the *poste-*

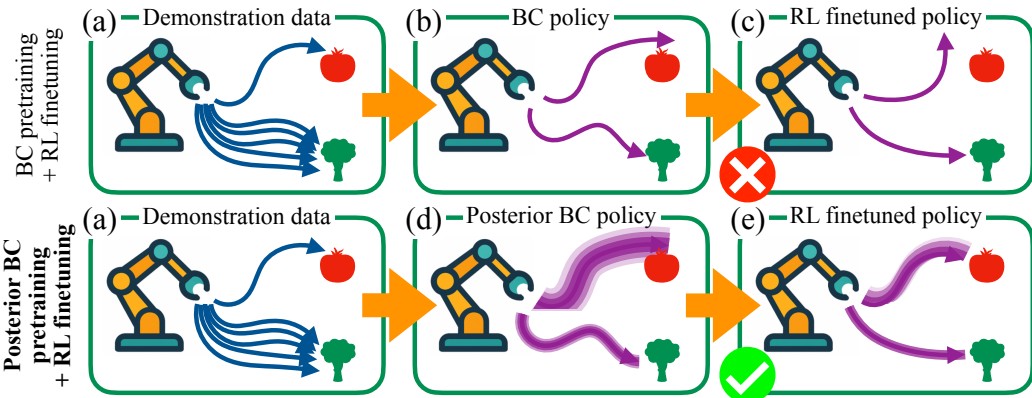

*Figure 1.* (a) We consider the setting where we are given demonstration data for some tasks of interest. (b) Standard BC pretraining fits the behaviors in the demonstrations, leading to effective performance in regions with high demonstration data density, yet can overcommit to the observed behaviors in regions with low data density. (c) This leads to ineffective RL finetuning, since rollouts from the BC policy provide little meaningful reward signal in such low data density regions, which is typically necessary to enable effective improvement. (d) In contrast, we propose *posterior behavioral cloning* (POSTBC), which instead of directly mimicking the demonstrations, trains a generative policy to fit the *posterior distribution* of the demonstrator's behavior. This endows the pretrained policy with a wider distribution of actions in regions of low demonstrator data density, while in regions of high data density it reduces to approximately the standard BC policy. (e) This wider action distribution in low data density regions allows for collection of diverse observations with more informative reward signal, enabling more effective RL finetuning, while in regions of high data density performance converges to that of the demonstrator.

*rior* distribution over the demonstrator's behavior. That is, assuming a uniform prior over the demonstrator's behavior and viewing the demonstration data as samples from the demonstrator's behavioral distribution, we seek to train a policy that models the posterior distribution of the demonstrator's behavior given these observations. This enables the pretrained policy to take into account its potential uncertainty about the demonstrator's behavior, and adjust the entropy of its action distribution based on this uncertainty. In states where it is uncertain about the demonstrator's actions, POSTBC samples from a high-entropy distribution, allowing for a more diverse set of actions that may enable further policy improvement, while in states where it is certain about the demonstrator's actions, it samples from a low-entropy distribution, simply mimicking what it knows to be the (correct) demonstrator behavior (see Figure 1).

We show that POSTBC leads to provable improvements over standard BC in terms of the potential for downstream RL performance. In particular, we focus on the ability of the pretrained policy to cover the demonstrator policy's actions—whether it samples all actions the demonstrator policy might sample—which, for finetuning approaches that rely on rolling out the pretrained policy, is a prerequisite to ensure finetuning can match the performance of the demonstrator. We show standard BC can provably fail to cover the demonstrator's distribution, while POSTBC *does* cover the demonstrator's distribution, incurs no suboptimality in the performance of the pretrained policy as compared to the BC policy, and achieves a near-optimal sampling cost out of all policy estimators which have pretrained performance no worse than the BC policy's.

Inspired by this, we develop a practical approach to approximating the posterior of the demonstrator in continuous action domains, and instantiate POSTBC with modern generative models—diffusion models—on robotic control tasks. Our instantiation relies only on pretraining with scalable supervised learning objectives—no RL is required in pretraining—and can be incorporated into existing BC training pipelines with minimal modification. We demonstrate experimentally that POSTBC pretraining can lead to significant performance gains in terms of the efficiency and effectiveness of RL finetuning, as compared to running RL finetuning on a policy pretrained with standard BC, and achieves these gains without decreasing the performance of the pretrained policy itself. We show that this holds for a variety of finetuning algorithms—both policy-gradient-style algorithms, and algorithms which explicitly refine or filter the distribution of the pretrained policy—enabling effective RL finetuning across a variety of challenging robotic tasks in both simulation and the real world.

## 2. Related Work

We highlight the most relevant work here. Please see Section A for discussion on other pretraining approaches (imitation learning, meta-learning, and reinforcement learning), methods for RL finetuning, and posterior sampling.

**BC pretraining.** BC training of expressive generative models—where the model is trained to predict the next "action" of the demonstrator—forms the backbone of pretraining for LLMs (Radford et al., 2018) and robotic control (Bojarski, 2016; Zhang et al., 2018; Rahmatizadeh et al., 2018; Stepputtis et al., 2020; Shafiullah et al., 2022; Gu

et al., 2023; Team et al., 2024; Zhao et al., 2024; Black et al., 2024; Kim et al., 2024). Experimentally, we focus in particular on policies parameterized as diffusion models (Sohl-Dickstein et al., 2015; Ho et al., 2020; Song et al., 2020), which have seen much attention in the robotics community (Chi et al., 2023; Ankile et al., 2024a; Zhao et al., 2024; Ze et al., 2024; Sridhar et al., 2024; Dasari et al., 2024; Team et al., 2024; Black et al., 2024; Bjorck et al., 2025), yet our approach can extend to other generative model classes as well. These works, however, simply pretrain with standard BC, and do not consider how the pretraining may affect RL finetuning performance.

**Pretraining for downstream finetuning.** Several recent works in the language domain aim to understand the relationship between pretraining and downstream finetuning (Springer et al., 2025; Zeng et al., 2025; Chen et al., 2025b; Jin et al., 2025; Chen et al., 2025a). A common thread through these works is that cross entropy loss is not predictive of downstream finetuning performance, and, in fact, low cross entropy loss can be anti-correlated with finetuning performance as the model can become *overconfident*. Most related to our work is the concurrent work of Chen et al. (2025a), which consider a notion of *coverage* closely related to our notion of *demonstrator action coverage*, and show that coverage generalizes faster than cross-entropy, and is an effective predictor of the downstream success of Best-of-$N$ sampling. While both our work and Chen et al. (2025a) focus on notions of coverage to enable downstream improvement, we see this work as complementary. Chen et al. (2025a) show coverage generalizes faster than cross-entropy, yet our results show BC pretraining can *still* fail to ensure meaningful coverage. Furthermore, Chen et al. (2025a) does not consider the tradeoff between policy performance and coverage that is a primary focus of our work, and their proposed pretraining intervention—gradient normalization—would not resolve the shortcomings of BC in our setting. Finally, Chen et al. (2025a) is primarily a theoretical work and focuses on discrete next-token prediction (indeed, all works cited above consider only discrete next-token prediction); in contrast, a primary focus of our work is on continuous control, and we demonstrate our approach scales to real-world robotic settings.

## 3. Preliminaries

**Mathematical notation.** Let $\lesssim$ denote inequality up to absolute constants, $\triangle_{\mathcal{X}}$ the simplex over $\mathcal{X}$, and $\mathrm{unif}(\mathcal{X})$ the uniform distribution over $\mathcal{X}$. $\mathbb{I}\{\cdot\}$ denotes the indicator function, $\mathbb{E}^{\pi}[\cdot]$ the expectation under policy $\pi$ and $\mathbb{E}[\cdot]$ the expectation over the demonstrator dataset.

**Markov decision processes.** We consider decision-making in the context of episodic, fixed-horizon Markov decision processes (MDPs). An MDP $\mathcal{M}$ is denoted by

a tuple $(\mathcal{S}, \mathcal{A}, \{P_h\}_{h=1}^{H}, P_0, r, H)$, where $\mathcal{S}$ is the set of states, $\mathcal{A}$ the set of actions, $P_h : \mathcal{S} \times \mathcal{A} \to \triangle_{\mathcal{S}}$ the next-state distribution at step $h$, $P_0 \in \triangle_{\mathcal{S}}$ the initial state distribution, $r_h : \mathcal{S} \times \mathcal{A} \to \triangle_{[0,1]}$ the reward distribution, and $H$ the horizon. Interaction with $\mathcal{M}$ proceeds in episodes of length $H$. At step 1, we sample a state $s_1 \sim P_0$, take an action $a_1 \in \mathcal{A}$, receive reward $r_1(s_1, a_1)$, and transition to state $s_2 \sim P_1(\cdot \mid s_1, a_1)$. This continues for $H$ steps until the MDP resets. We let $\mathcal{J}(\pi) := \mathbb{E}^{\pi}[\sum_{h=1}^{H} r_h(s_h, a_h)]$ denote the expected reward for policy $\pi$. In general, our goal is to find a policy that maximizes $\mathcal{J}(\pi)$.

**Behavioral cloning.** We assume we are given some dataset $\mathfrak{D} = \{(s_1^t, a_1^t, \ldots, s_H^t, a_H^t)\}_{t=1}^{T}$ collected by running a *demonstrator* policy $\pi^{\beta}$ on $\mathcal{M}$, so that $(s_1^t, a_1^t, \ldots, s_H^t, a_H^t)$ denotes a trajectory rollout of $\pi^{\beta}$, with $a_h^t \sim \pi_h^{\beta}(\cdot \mid s_h^t)$. We assume $\pi^{\beta}$ is Markovian but otherwise make no further assumptions on it (so, in particular, $\pi^{\beta}$ may be stochastic and suboptimal). Our demonstrator dataset does not include reward labels—preventing standard offline RL approaches from applying—but we assume access to reward labels during online interactions.

*Behavioral cloning* (BC) attempts to fit a policy $\widehat{\pi}^{\mathrm{bc}}$ to match the distribution of actions observed in $\mathfrak{D}$. Typically this is achieved via supervised learning, where $\widehat{\pi}^{\mathrm{bc}}$ is trained to predict $a$ given $s$ for $(s, a) \in \mathfrak{D}$. In the tabular setting, which we consider in Section 4, the natural choice for $\widehat{\pi}^{\mathrm{bc}}$ models the empirical action distribution in $\mathfrak{D}$:

$$
\begin{aligned}
\widehat{\pi}_h^{\mathrm{bc}}(a \mid s) := \tfrac{T_h(s,a)}{T_h(s)} \cdot \mathbb{I}\{T_h(s) > 0\} \\
+ \mathrm{unif}(\mathcal{A}) \cdot \mathbb{I}\{T_h(s) = 0\}
\end{aligned} \tag{1}
$$

where $T_h(s, a) = \sum_{t=1}^{T} \mathbb{I}\{(s_h^t, a_h^t) = (s, a)\}$ and $T_h(s) = \sum_{t=1}^{T} \mathbb{I}\{s_h^t = s\}$. The following result bounds the suboptimality of this estimator, and shows that it is optimal.

**Proposition 1** (Rajaraman et al. (2020)). *If $\mathfrak{D}$ contains $T$ demonstrator trajectories, we have $\mathcal{J}(\pi^{\beta}) - \mathbb{E}[\mathcal{J}(\widehat{\pi}^{\mathrm{bc}})] \lesssim \frac{H^2 S \log T}{T}$. Furthermore, for any estimator $\widehat{\pi}$, there exists some MDP $\mathcal{M}$ and demonstrator $\pi^{\beta}$ such that $\mathcal{J}(\pi^{\beta}) - \mathbb{E}[\mathcal{J}(\widehat{\pi})] \gtrsim \min\{H, \frac{H^2 S}{T}\}$.*

In other words, without additional reward information, we cannot in general hope to obtain a policy from $\mathfrak{D}$ that does better than (1), in terms of the reward collected by the pretrained policy. Note that the estimator (1) is, under the uniform demonstrator prior (i.e. the demonstrator is equally likely to play each action in each state), the *maximum a posterior* (MAP) estimate of the demonstrator's behavior.

## 4. Achieving Demonstrator Action Coverage via Posterior Sampling

In this section we seek to understand how pretraining affects the ability to further improve the downstream policy

with RL finetuning, and how we might pretrain to enable downstream improvement. For simplicity, in this section we assume $\mathcal{M}$ is tabular, and let $S$ and $A$ denote the cardinalities of the state and action spaces, respectively.

### 4.1. Demonstrator Action Coverage

The performance of RL finetuning depends significantly on the RL algorithm applied. Rather than limiting our results to a particular RL algorithm, we instead focus on what is often a prerequisite for effective application of any such approach—demonstrating that the *support* of the pretrained policy is sufficient to enable improvement. In particular, we consider the following definition for the "effective" support of a policy, relative to the demonstrator policy $\pi^\beta$.

**Definition 4.1** (Demonstrator Action Coverage)**.** We say policy $\pi$ achieves demonstrator action coverage with parameter $\gamma > 0$ if, for all $(s, h) \in \mathcal{S} \times [H]$ and $a \in \mathcal{A}$, we have $\pi_h(a \mid s) \geq \gamma \cdot \pi_h^\beta(a \mid s)$.

The majority of RL finetuning approaches rely on rolling out the pretrained policy—which we denote as $\widehat{\pi}^{\mathrm{pt}}$—online, and using the collected observations to finetune its behavior. If our pretrained policy achieves demonstrator action coverage with parameter $\gamma$, then this ensures that any action sampled by $\pi^\beta$ will also be sampled by $\widehat{\pi}^{\mathrm{pt}}$ in these rollouts (with some probability). While this is not a *sufficient* condition for online improvement, it is a *necessary* condition for performing as well as the demonstrator $\pi^\beta$ (as Proposition 2 shows), and is therefore also a necessary condition for improving over $\pi^\beta$. Furthermore, the *value* of $\gamma$ has impact on the cost of RL finetuning. A policy $\pi$ which achieves demonstrator action coverage with parameter $\gamma$ requires a factor of $1/\gamma$ more samples than $\pi^\beta$ to ensure it samples some action in the support of $\pi^\beta$. For approaches such as Best-of-$N$ sampling that rely on sampling many actions from the pretrained policy and then taking the best one, a large value of $\gamma$ ensures we can efficiently sample actions likely to be sampled by the demonstrator policy $\pi^\beta$, while if $\gamma$ is small, it may take a significant number of samples to sample an action necessary for improvement.

**Problem Statement: Demonstrator Action Coverage with BC-Pretrained Performance.** In the following, we aim to understand how we can pretrain policies that (a) achieve demonstrator action coverage with values of $\gamma$ as large as possible and (b) have the same pretrained performance as the BC-pretrained performance.

### 4.2. BC Fails to Achieve Demonstrator Action Coverage

We first consider standard BC, i.e. (1). The following result shows that the estimator in (1), despite achieving the best possible suboptimality rate, can fail to achieve a meaningful guarantee on demonstrator action coverage.

**Proposition 2.** *Fix* $\epsilon \in (0, 1/8)$. *Then there exist some MDPs* $\mathcal{M}^1, \mathcal{M}^2$ *and demonstrator policy* $\pi^\beta$ *such that, if* $\mathcal{M} \in \{\mathcal{M}^1, \mathcal{M}^2\}$, *unless* $T \geq \frac{1}{20\epsilon}$, *we have that, with probability at least* $1/2$: $\mathcal{J}(\pi^\beta) - \epsilon > \max_{\pi \in \widehat{\Pi}} \mathcal{J}(\pi)$, *for* $\widehat{\Pi} := \{\pi : \pi_h(a \mid s) = 0 \text{ if } \widehat{\pi}_h^{\mathrm{bc}}(a \mid s) = 0, \forall s, a, h\}$. *Furthermore, for any* $T' > 0$,

$$\min_{\widehat{\pi}^{T'}} \max_{i \in \{1,2\}} \mathbb{E}^{\mathcal{M}^i, \widehat{\pi}^{\mathrm{bc}}}[\max_\pi \mathcal{J}^{\mathcal{M}^i}(\pi) - \mathcal{J}^{\mathcal{M}^i}(\widehat{\pi}^{T'})] \geq \tfrac{1}{2},$$

*where* $\mathbb{E}^{\mathcal{M}^i, \widehat{\pi}^{\mathrm{bc}}}[\cdot]$ *denotes the expectation over trajectories generated by rolling out* $\widehat{\pi}^{\mathrm{bc}}$ *on* $\mathcal{M}^i$, *and* $\widehat{\pi}^{T'}$ *is a policy estimator obtained after* $T'$ *such rollouts.*

Proposition 2 shows that, unless we have a sufficiently large demonstration dataset, half of the time (i.e. half of the random draws of the demonstrator dataset) the policy returned by standard BC will not contain a near-optimal policy in its support and, furthermore, rolling out $\widehat{\pi}^{\mathrm{bc}}$ on $\mathcal{M}$ will not allow us to learn a near-optimal policy. This also shows that demonstrator action coverage is, in some cases, a necessary condition for RL improvement—without it, we will not sample actions played by the demonstrator, and will be unable to determine which lead to the best performance.

A straightforward solution to this shortcoming of BC is to simply add exploration noise to our pretrained policy—rather than playing $\widehat{\pi}^{\mathrm{bc}}$ at every step, with some probability play a random action. While this will clearly ensure demonstrator action coverage is achieved, the following result shows there is a fundamental tradeoff between policy suboptimality and the value of $\gamma$.

**Proposition 3.** *Fix* $T > 0$, $H \geq 2$, $S \geq \lceil \log_2 4T \rceil + 2$, $\xi \geq 0$, *define* $\epsilon := \frac{H^2 S \log T}{T} + \xi$, *and assume* $\epsilon \leq \frac{1}{2}$. *Define the policy* $\widehat{\pi}^{\mathrm{u},\alpha}$ *as* $\widehat{\pi}_h^{\mathrm{u},\alpha}(\cdot \mid s) := (1 - \alpha) \cdot \widehat{\pi}_h^{\mathrm{bc}}(\cdot \mid s) + \alpha \cdot \mathrm{unif}(\mathcal{A})$. *Then there exists some MDP* $\mathcal{M}$ *with* $S$ *states, 2 actions, and horizon* $H$ *where, in order to ensure:*

1. $\mathcal{J}(\pi^\beta) - \mathbb{E}[\mathcal{J}(\widehat{\pi}^{\mathrm{u},\alpha})] \leq \epsilon$,

2. $\widehat{\pi}^{\mathrm{u},\alpha}$ *achieves demonstrator action coverage with parameter* $\gamma$ *and probability* $1 - \delta$, *for* $\delta \in (0, 1/4e)$,

*we must have* $\alpha \leq 32\epsilon$ *and* $\gamma \leq \frac{64}{A} \cdot \epsilon$. *Furthermore, with probability at least* $1/4e$, $\mathcal{J}(\pi^\beta) - \frac{1}{T} \cdot \epsilon > \max_{\pi \in \widehat{\Pi}} \mathcal{J}(\pi)$.

In order to achieve the $\frac{H^2 S \log T}{T}$ suboptimality rate achieved by standard BC, Proposition 3 shows that we can only guarantee demonstrator action coverage with $\gamma \lesssim \frac{1}{A} \cdot \frac{H^2 S \log T}{T}$. In other words, to ensure we sample a particular action from $\widehat{\pi}^{\mathrm{u},\alpha}$ that is sampled by $\pi^\beta$, it will require sampling a factor of $\frac{AT}{H^2 S \log T}$ *more* samples from $\widehat{\pi}^{\mathrm{u},\alpha}$ than it would require from $\pi^\beta$, which could be a significant cost when $T$ is large. Furthermore, Proposition 3 shows that this limitation is critical—if we seek to shortcut this exploration and set $\alpha \leftarrow 0$, we will fail to match the performance of $\pi^\beta$ completely.

### 4.3. Posterior Demonstrator Policy Achieves Demonstrator Action Coverage

Next we show that a mixture of the standard BC policy and the *posterior* on the demonstrator's policy achieves a near optimal balance between policy suboptimality and demonstrator action coverage.

**Definition 4.2** (Posterior Demonstrator Policy)**.** Given prior distribution $P^\beta_{\text{prior}} \in \triangle_\Pi$ over demonstrator policies, let $P^\beta_{\text{post}}(\cdot \mid \mathfrak{D})$ denote the posterior distribution given demonstration dataset $\mathfrak{D}$. We then define the *posterior demonstrator policy* $\widehat{\pi}^{\text{post}}$ as $\widehat{\pi}^{\text{post}}_h(a \mid s) := \mathbb{E}_{\pi \sim P^\beta_{\text{post}}(\cdot | \mathfrak{D})}[\pi_h(a \mid s)]$.

$\widehat{\pi}^{\text{post}}$ is therefore the expected posterior policy of the demonstrator under prior $P^\beta_{\text{prior}}$ given observations $\mathfrak{D}$. Critically, this takes into account the entire posterior distribution of the demonstrator's behavior, in contrast to the MAP estimate produced by standard BC, which simply returns a point estimate of the behavior. In practice, we require a slightly regularized version of $\widehat{\pi}^{\text{post}}$, $\widehat{\pi}^{\text{post},\lambda}$, which is identical to $\widehat{\pi}^{\text{post}}$ if $HT \lesssim e^A$, and otherwise adds a small amount of additional regularization (see Section B.3 for a precise definition). We have the following.

**Theorem 1.** *Let $P^\beta_{\text{prior}}$ be the uniform distribution over Markovian policies, and set $\widehat{\pi}^{\text{pt}}$ to*

$$\widehat{\pi}^{\text{pt}}_h(a|s) = (1-\alpha) \cdot \widehat{\pi}^{\text{bc}}_h(a|s) + \alpha \cdot \widehat{\pi}^{\text{post},\lambda}_h(a|s) \quad (2)$$

*for $\alpha = \frac{1}{\max\{A, H, \log(HT)\}}$. Then*

$$\mathcal{J}(\pi^\beta) - \mathbb{E}[\mathcal{J}(\widehat{\pi}^{\text{pt}})] \lesssim \frac{H^2 S \log T}{T},$$

*and with probability at least $1 - \delta$, for all $(s, a, h)$,*

$$\widehat{\pi}^{\text{pt}}_h(a \mid s) \gtrsim \frac{1}{A+H+\log(HT)} \cdot \min\left\{ \frac{\pi^\beta_h(a|s)}{\log(SH/\delta)}, \frac{1}{A+\log(HT)} \right\}.$$

Theorem 1 shows that by setting $\widehat{\pi}^{\text{pt}}$ to a mixture of the BC policy and the posterior demonstrator policy, we obtain the same suboptimality guarantee as standard BC. Furthermore, this policy achieves demonstrator action coverage with $\gamma \approx 1/(A + H)$, only requiring a factor of $\approx A + H$ more samples to ensure we sample a particular action from $\pi^\beta$ than $\pi^\beta$. We refer to this approach as *posterior behavioral cloning* (POSTBC). The following result shows that the scaling in $\gamma$ POSTBC achieves is nearly unimprovable.

**Theorem 2.** *Fix any $A > 1$ and $T > 1$. Then there exists a family of MDPs $\{\mathcal{M}^i\}_{i \in [A]}$ such that each $\mathcal{M}^i$ has $A$ actions and $S = H = 1$, and if any estimator $\widehat{\pi}$ satisfies $\mathcal{J}^{\mathcal{M}^i}(\pi^{\beta,i}) - \mathbb{E}^{\mathcal{M}^i}[\mathcal{J}(\widehat{\pi})] \leq c \cdot \frac{H^2 S \log T}{T}$ for all $i \in [A]$ and some constant $c > 0$, then for $\widehat{\pi}$ to achieve demonstrator action coverage with respect to $\pi^{\beta,i}$ on each $\mathcal{M}^i$ with probability at least $\delta \in (0, 1/4]$, we must have $\gamma \leq c \cdot \frac{\log T}{A}$.*

Theorem 2 shows that, to match the suboptimality guarantee of the BC policy, no estimator can achieve demonstrator action coverage with $\gamma$ larger than $\approx 1/A$. Thus, the demonstrator action coverage achieved by POSTBC is nearly unimprovable, matching the lower bound as long as $H \leq A$. In other words, if we want a policy that preserves the optimality of $\widehat{\pi}^{\text{bc}}$ while playing a diverse enough action distribution to enable further online improvement, mixing the posterior demonstrator policy with the BC policy achieves a near-optimal tradeoff. This is in contrast to the BC policy, which does not achieve a meaningful guarantee on demonstrator action coverage, as well as the BC policy augmented with random exploration, which in order to match the suboptimality of the BC policy achieves a very suboptimal guarantee on demonstrator action coverage.

## 5. Practical Posterior Behavioral Cloning

Next, we show how POSTBC can be instantiated in continuous control settings using expressive generative models.

### 5.1. Sampling from the Posterior Demonstrator Policy for Gaussian Demonstrators

To motivate our practical instantiation, assume:

$$\pi^\beta_h(\cdot \mid s) = \mathcal{N}(\mu_h(s), \sigma^2_h(s) \cdot I),$$

for some (unknown) $\mu_h(s) \in \mathbb{R}^d$ and (known) $\sigma_h(s) \in \mathbb{R}$. Assume we have observations $\mathfrak{D} = \{a_1, \ldots, a_T\} \sim \pi^\beta_h(\cdot \mid s)$, and a $\mathcal{N}(0, I)$ prior on $\mu_h(s)$. Our theory suggests that instead of fitting the BC policy, we should fit the posterior demonstrator policy $\widehat{\pi}^{\text{post}}$. In this Gaussian setting, it is straightforward to show that $\widehat{\pi}^{\text{post}}_h(\cdot \mid s)$ is the distribution:

$$\mathcal{N}\left( \frac{1}{\sigma^2_h(s)+T} \cdot \sum_{t=1}^T a_t, \frac{\sigma^2_h(s)}{\sigma^2_h(s)+T} \cdot I + \sigma^2_h(s) \cdot I \right).$$

While in the Gaussian setting we can easily sample from this distribution, we wish to motivate a generalizable procedure that extends to settings where sampling is less straightforward. To this end, we first note that the BC policy (the MAP estimator) is simply the distribution $\mathcal{N}(\frac{1}{\sigma^2_h(s)+T} \cdot \sum_{t=1}^T a_t, \sigma^2_h(s) \cdot I)$. To generate a sample from $\widehat{\pi}^{\text{post}}_h(\cdot \mid s)$, it then suffices to sample from the BC policy and perturb the sample by noise $w \sim \mathcal{N}(0, \frac{\sigma^2_h(s)}{\sigma^2_h(s)+T} \cdot I)$. The following result, an extension of Osband et al. (2018), shows that there is a close connection between this noise distribution and the posterior on $\mu_h(s)$, and that we can generate a sample from the posterior on $\mu_h(s)$ with a simple optimization procedure.

**Proposition 4.** *We have $P^\beta_{\text{post}}(\cdot \mid \mathfrak{D}) = \mathcal{N}(\frac{1}{\sigma^2_h(s)+T} \cdot \sum_{t=1}^T a_t, \frac{\sigma^2_h(s)}{\sigma^2_h(s)+T} \cdot I)$ and, if we set*

$$\widehat{\mu}_h(s) = \arg\min_\mu \sum_{i=1}^T \|\mu - \widetilde{a}_i\|^2_2 + \sigma^2_h(s) \cdot \|\mu - \widetilde{\mu}_h(s)\|^2_2,$$

*for $\widetilde{a}_t = a_t + w_t$, $w_t \sim \mathcal{N}(0, \sigma_h^2(s) \cdot I)$, and $\widetilde{\mu}_h(s) \sim \mathcal{N}(0, I)$, then $\widehat{\mu}_h(s) \sim P_{\mathrm{post}}^\beta(\cdot \mid \mathfrak{D})$.*

Thus, to generate a sample $w \sim \mathcal{N}(0, \frac{\sigma_h^2(s)}{\sigma_h^2(s)+T} \cdot I)$, we can first generate samples $\widehat{\mu}_h(s)$, compute their empirical variance, $\mathrm{cov}_h(s)$, and sample from a Gaussian with mean 0 and variance $\mathrm{cov}_h(s)$. Altogether, we have the following procedure to sample $\widehat{a} \sim \widehat{\pi}_h^{\mathrm{post}}(\cdot \mid s)$:

1. Compute the BC policy $\widehat{\pi}^{\mathrm{bc}}$ from observations $\mathfrak{D}$.

2. Compute samples from the posterior $P_{\mathrm{post}}^\beta$ using the optimization procedure of Proposition 4 and estimate the variance $\mathrm{cov}_h(s)$ of the posterior from these samples.

3. Generate samples $\widetilde{a} \sim \widehat{\pi}_h^{\mathrm{bc}}(\cdot \mid s)$ and $w \sim \mathcal{N}(0, \mathrm{cov}_h(s))$, and set $\widehat{a} \leftarrow \widetilde{a} + w$.

While in the Gaussian setting simpler methods would suffice to sample from $\widehat{\pi}^{\mathrm{post}}$, each step in this procedure can be easily extended to more complex settings, suggesting a generalizable approach to approximate samples from $\widehat{\pi}^{\mathrm{post}}$.

### 5.2. Practical Instantiation of POSTBC

In practice our data likely does not satisfy this Gaussianity assumption, and we also wish to incorporate function approximation. To accommodate this, we seek to approximate the procedure outlined above in more general settings, and summarize our approach in Algorithm 1.

---

**Algorithm 1** Posterior Behavioral Cloning (POSTBC)

---

1: **input:** demonstration dataset $\mathfrak{D}$, generative model class $\widehat{\pi}^\theta$, ensemble size $K$, posterior weight $\alpha$
2: Fit $K$ predictors $f_\ell$ to perturbed $\mathfrak{D}$ (see Algorithm 2)
3: Set $\mathrm{cov}(\cdot)$ to covariance of $\{f_\ell\}_{\ell \in [K]}$
4: **for** $i = 1, 2, 3, \ldots$ **do**
5:     Sample batch $\mathfrak{B}_i \sim \mathrm{unif}(\mathfrak{D})$
6:     For all $(s, a) \in \mathfrak{B}_i$, sample $w_s \sim \mathcal{N}(0, \mathrm{cov}(s))$, and set $\widetilde{\mathfrak{B}}_i \leftarrow \{(s, a + \alpha \cdot w_s) : s \in \mathfrak{B}_i\}$
7:     Take gradient step on $\widehat{\pi}^\theta$ for loss computed on $\widetilde{\mathfrak{B}}_i$
8: **return** $\widehat{\pi}^\theta$

---

Algorithm 1 first trains an ensemble of predictors $\{f_\ell\}_{\ell \in [K]}$ on perturbed versions of $\mathfrak{D}$, which it then uses to approximate the posterior covariance as in step 2 above. It then trains a generative policy to fit the demonstrations in $\mathfrak{D}$, perturbed by noise of covariance $\mathrm{cov}(\cdot)$, approximating step 3 above. This produces a single policy that approximates the posterior mixture in Equation (2), modeling the posterior demonstrator policy and, Theorem 1 suggests, leading to a more effective initialization for RL finetuning, instantiating the behavior illustrated in Figure 1.

Implementing Algorithm 1 only requires minor modification to standard generative policy training (simply train an

initial ensemble of predictors, then add noise to the action target for each batch sampled), and in particular does not require RL in pretraining. While any expressive generative model class can be used for $\widehat{\pi}^\theta$, for all the following experiments we utilize a diffusion model. Please see Section D for further details on the practical instantiation of POSTBC.

## 6. Experiments

Finally, we seek to demonstrate that in practice POSTBC (a) enables more efficient RL finetuning of pretrained policies, and (b) produces a pretrained policy that itself performs effectively, on par with the BC pretrained policy. We focus on continuous control domains, in particular robotic control. We test on both the `Robomimic` (Mandlekar et al., 2021) and `Libero` (Liu et al., 2023) simulators, as well as a real-world WidowX 250 6-DoF robot arm. For `Robomimic`, we consider pretraining single-task policies on the provided set of demonstrations, in particular for the `Lift`, `Can`, and `Square` tasks. `Libero` contains a diverse set of robot manipulation tasks, and we focus in particular on the 16 tasks from `Kitchen Scene 1-3` from the `Libero 90` suite. We pretrain a single policy on the provided demonstrations across these tasks, then use this policy as the starting point for RL finetuning, and run single-task RL on the tasks from `Kitchen Scene 1-3`. In all cases, we instantiate $\widehat{\pi}^{\mathrm{pt}}$ with a diffusion model, a standard parameterization for BC policies in robotic control (Chi et al., 2023; Dasari et al., 2024; Team et al., 2024). In all cases, we use a binary success reward for RL finetuning. See Figures 5 and 6 for visualizations of our settings.

In principle, POSTBC can be combined with any RL finetuning algorithm, and we seek to demonstrate that it improves performance on a representative set of approaches. In particular, we consider DSRL (Wagenmaker

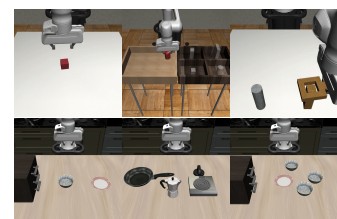

*Figure 5.* `Robomimic` and `Libero` settings.

et al., 2025a), which refines a pretrained diffusion policy's distribution by running RL over its latent-noise space, DPPO (Ren et al., 2024), an on-policy policy-gradient-style algorithm for finetuning diffusion policies, and Best-of-$N$ sampling. Best-of-$N$ can be instantiated in a variety of ways (see e.g. Chen et al. (2022); Hansen-Estruch et al. (2023); He et al. (2024); Nakamoto et al. (2024); Dong et al. (2025b)). Here we instantiate it by rolling out the pretrained policy on the task of interest $T_{\mathrm{on}}$ times (where $T_{\mathrm{on}}$ is specified in our results) to collect trajectories labeled with reward, and train a $Q$-function via IQL (Kostrikov et al., 2021) on these trajectories. At test time, we again roll out the pretrained policy but at each state sample $N$

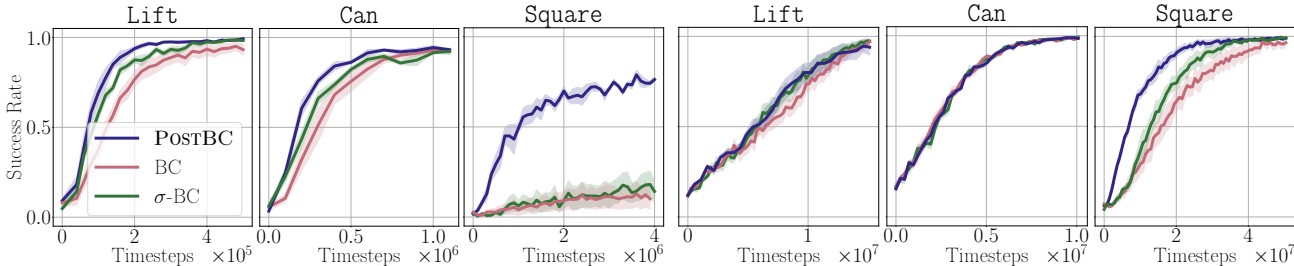

*Figure 2.* Comparison of DSRL finetuning performance combined with different BC pretraining approaches on Robomimic.

*Figure 3.* Comparison of DPPO finetuning performance combined with different BC pretraining approaches on Robomimic.

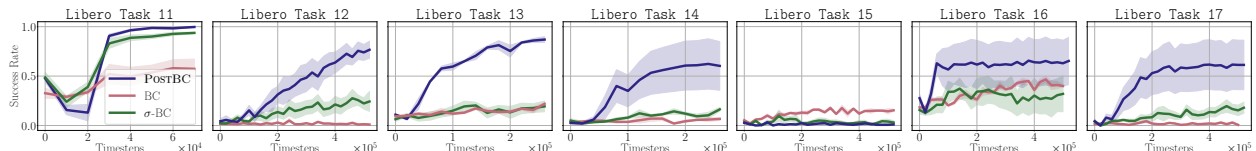

*Figure 4.* Comparison of DSRL finetuning performance on all tasks from Libero 90, Kitchen Scene 2.

actions from the policy, and play the action that has the largest value under this $Q$-function.

As baselines, we consider running standard BC pretraining on $\mathfrak{D}$, as well as what we refer to as $\sigma$-BC, where instead of perturbing the actions in $\mathfrak{D}$ by the posterior variance as in Algorithm 1, we instead perturb them by uniform, state-independent noise with variance $\sigma^2$. This is then equivalent to POSTBC, except we set $\text{cov}(s) = \sigma^2 \cdot I$ for some fixed $\sigma > 0$ in Algorithm 1 (note that this is a continuous analog to the approach considered in Proposition 3). This itself is a novel approach and our theory predicts it too may lead to improved performance over pretraining with standard BC. On Robomimic, we also compare against VALUEDICE (Kostrikov et al., 2019) (which we abbreviate as DICE), as a representative non-BC imitation learning approach. Rather than training to match the demonstrator's actions via supervised learning, VALUEDICE attempts to learn a policy with state distribution matching the state distribution of the demonstrations, and only requires access to offline demonstration data. All Robomimic results are averaged over 5 seeds, and Libero results are averaged over 3 seeds. For all experiments, error bars denote 1 standard error. See Section D for additional experimental details.

### 6.1. POSTBC Enables Efficient RL Finetuning

Our results from running DSRL on Robomimic are given in Figure 2 and on Libero in Figure 4. On Robomimic, POSTBC significantly outperforms both baselines on Square, and on Lift and Can requires roughly $2\times$ fewer samples to achieve 75% success than BC. For Libero, we run DSRL on all tasks from Kitchen Scene 2. We see that POSTBC pretraining leads to significant gains for Libero, enabling efficient RL finetuning in settings where both standard BC pretraining and $\sigma$-BC pretraining fail. In particular, on each task other

than Task 15, POSTBC leads to large gains—we hypothesis that on Task 15, where all methods fail to learn, this is due primarily to the poor pretrained policy performance for all methods, which often makes RL improvement challenging. Our Best-of-$N$ results are given in Table 1. Across settings, POSTBC-pretraining leads to consistent improvements over both BC- and $\sigma$-BC-pretrained policies for Best-of-$N$, and also consistently outperforms VALUEDICE.

Our results for DPPO are given in Figure 3 where we see that POSTBC pretraining again leads to substantial gains on Square (approximately $2\times$ fewer samples to reach 75% performance compared to BC). This illustrates that POSTBC still improves performance even for RL finetuning algorithms that modify the weights of the pretrained policy, and for which the resulting actions are therefore not explicitly constrained to the actions played by the pretrained policy. We hypothesize that this is due to the ability of POSTBC to adapt its exploration based on its certainty of demonstrator behavior. POSTBC naturally increases exploration in states where it is uncertain what the demonstrator will do—states where exploration is required—while ensuring it does not over-explore in states where it knows that the demonstrator will do and exploration is not required. This is true whether we play an algorithm that modifies the weights of the pretrained policy or not, so we would expect improvement even when running with DPPO. Together, these results illustrate that POSTBC enables more effective RL finetuning on both single-task (Robomimic) and multi-task (Libero) settings, and across state- and image-based observations.

### 6.2. POSTBC Preserves Pretrained Performance

We next consider POSTBC pretrained performance. We provide results on pretrained performance for Robomimic

| Task | Best-of-$N$ (1000 Rollouts) | | | | Best-of-$N$ (2000 Rollouts) | | | | Pretrained Performance | |
|---|---|---|---|---|---|---|---|---|---|---|
| | BC | $\sigma$-BC | DICE | POSTBC | BC | $\sigma$-BC | DICE | POSTBC | BC | POSTBC |
| Robomimic Lift | 59.4 ±2.2 | 61.5 ±3.9 | 44.4 ±7.7 | **74.2** ±3.0 | 68.1 ±2.2 | 76.1 ±3.5 | 47.6 ±7.8 | **81.3** ±5.7 | **71.0** ±0.5 | 69.7 ±1.5 |
| Robomimic Can | 70.3 ±1.7 | **77.9** ±1.2 | 17.0 ±5.7 | 75.0 ±2.5 | 76.9 ±1.5 | 82.4 ±1.6 | 41.8 ±8.4 | **84.5** ±1.3 | 43.1 ±0.9 | **44.7** ±1.0 |
| Robomimic Square | 44.8 ±0.7 | 48.1 ±2.2 | 6.9 ±0.9 | **52.4** ±1.9 | 54.4 ±1.3 | 54.2 ±3.7 | 8.3 ±1.3 | **56.8** ±3.2 | 17.9 ±0.7 | **18.1** ±0.8 |
| Libero Scene 1 (5 tasks) | 37.7 ±2.5 | 57.9 ±3.4 | - | **67.0** ±5.6 | 46.1 ±2.6 | 63.1 ±4.1 | - | **77.7** ±1.3 | 23.6 ±1.7 | **25.3** ±1.2 |
| Libero Scene 2 (7 tasks) | 21.5 ±1.2 | 26.9 ±1.0 | - | **42.0** ±2.2 | 23.9 ±0.7 | 29.0 ±1.6 | - | **49.5** ±2.9 | 11.4 ±0.2 | **12.3** ±1.5 |
| Libero Scene 3 (4 tasks) | 47.7 ±0.7 | 53.2 ±2.3 | - | **63.3** ±5.1 | 45.8 ±3.3 | 60.8 ±1.9 | - | **70.0** ±0.4 | 39.5 ±1.5 | **40.8** ±1.3 |
| Libero All (16 tasks) | 33.1 ±0.5 | 43.1 ±0.8 | - | **55.1** ±2.1 | 36.3 ±0.3 | 47.6 ±2.8 | - | **63.4** ±1.3 | 22.2 ±0.3 | **23.5** ±0.5 |

*Table 1.* Comparison of success rates of Best-of-$N$ sampling on `Robomimic` and `Libero`, for different pretraining approaches. Bolded text denotes best approach.

*Table 2.* Pretrained success on `Robomimic` and `Libero`.

| Task | Pretrained Performance | | Best-of-$N$ (100 Rollouts) | |
|---|---|---|---|---|
| | BC | POSTBC | BC | POSTBC |
| Pick up banana | 2/20 | **4/20** | 10/20 | **16/20** |
| Put corn in pot | 3/20 | **7/20** | 5/20 | **13/20** |

*Table 3.* Comparison of BC and POSTBC on real-world WidowX tasks with Best-of-$N$ sampling. POSTBC enables significantly larger improvement, while preserving pretrained performance.

| BC rollout + BC test | BC rollout + POSTBC test | POSTBC rollout + BC test | POSTBC rollout + POSTBC test |
|---|---|---|---|
| 68.1 ±2.2 | **82.7** ±1.9 | 29.4 ±3.6 | **81.3** ±5.7 |

*Table 4.* Best-of-$N$ sampling on `Robomimic Lift` with 2000 rollouts, varying the rollout policy used to collect data to train the $Q$-function, and the test-time ("test") policy.

and `Libero` in Table 2. As these results illustrate, across both single-task and multi-task settings, the POSTBC policy performs comparably to, or even better than, the BC policy in terms of pretrained performance. Please see Table 5 for pretrained performance of $\sigma$-BC and VALUEDICE; in general, these approaches achieve pretrained performance comparable to or somewhat worse than that of BC. Combined with the results of Section 6.1, this shows that POSTBC enables more effective RL finetuning without hurting the performance of the pretrained policy.

### 6.3. POSTBC Scales to Real-World Settings

We next show that POSTBC scales to real-world robotic settings, leading to improvement in real-world RL finetuning over standard BC pretraining. We evaluate on the WidowX 250 6-DoF robot arm and consider two tasks on the scene in Figure 6. We first collect 10 human teleoperation demonstrations for the task "`Put corn in pot`", where the objective is to pick up the corn and set it in the pot.

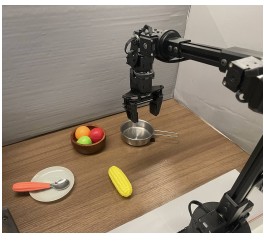

*Figure 6.* WidowX setup.

We train diffusion policies with standard BC as well as POSTBC on these demonstrations. For RL finetuning we consider two tasks—the original "`Put corn in pot`" task, and the task "`Pick up banana`" where the corn is replaced with a banana and the goal of the robot is to pick up the banana (see Figure 9). For RL finetuning, we consider the Best-of-$N$ procedure outlined above, and utilize 100 rollouts per task.

Our results are given in Table 3. We see that POSTBC leads to significant improvements over BC—in both tasks achieving significantly higher final success rate. In particular, for the "`Put corn in pot`" task, RL finetuning only marginally improves the performance of the BC policy, while the POSTBC policy leads to significant improve-

ment. Furthermore, POSTBC pretraining actually leads to improved pretrained performance compared to the BC policy. This illustrates that POSTBC scales to real robot settings, providing improved RL finetuning performance without decreasing pretrained performance.

### 6.4. Understanding POSTBC

Finally, we seek to provide insight into how POSTBC improves RL finetuning performance (please see Section D.4 for additional ablations). We first aim to disambiguate the role of the exploration a POSTBC policy may provide over a BC policy, versus the role that having access to a larger action distribution at test time might play. For Best-of-$N$ sampling, we can decouple this by selecting the rollout policy that collects data to learn the $Q$-function with IQL, and the policy whose actions we sample from and filter with the learned $Q$-function at test-time. We consider mixing the role of the BC and POSTBC policy on `Robomimic Lift` in Table 4. We find that using POSTBC as the test-time policy is critical to achieving effective performance, but that this performance is achievable whether we use BC or POSTBC for the rollout policy. This suggests that the utility of POSTBC is primarily in its ability to provide a wider range of actions that can be sampled at test time.

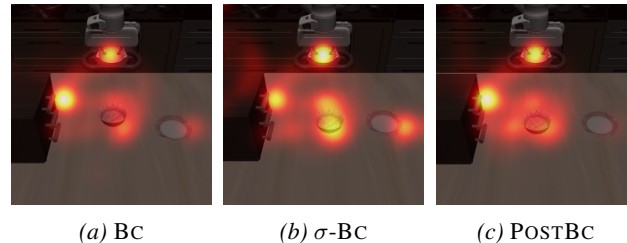

*(a)* BC       *(b)* $\sigma$-BC       *(c)* POSTBC

*Figure 7.* Qualitative analysis of POSTBC on Libero.

Next we consider the qualitative behavior of the POSTBC pretrained policy compared to the BC and $\sigma$-BC policies.

We illustrate this on Libero task "`Open the top drawer of the cabinet and put the bowl in it`" and display a heatmap of the visitations for each policy in Figure 7 (please see Section D.4 for visualizations on additional tasks). We see that POSTBC exhibits the widest distribution of states around the bowl, ensuring that it covers the behaviors necessary to reliably pick up the bowl, the most challenging aspect of this task. In contrast, BC and $\sigma$-BC exhibit less diversity around the bowl, giving them only a narrow set of behaviors to finetune from. At the same time, while exhibiting diversity around the states relevant to the task, POSTBC focuses its behavior only on these relevant behaviors, ensuring the pretrained policy still performs effectively. In contrast, $\sigma$-BC also interacts with the plate, which is irrelevant to this task and would therefore hurt its pretrained performance.

## 7. Conclusion

In this work, we have proposed a novel approach to pretraining policies from demonstrations that ensures the pretrained performance is no worse than that of the BC policy, while expanding the action distribution to enable more effective RL finetuning. We have shown that this approach does indeed lead to improved RL finetuning performance in practice, scaling to real-world robotic settings. We believe this work motivates a variety of interesting questions for future work.

- Our demonstrator action coverage condition introduced in Section 4.1 is a *necessary* condition, in some cases, for RL finetuning to reach the performance of the demonstrator policy, as Proposition 2 shows. In general, however, demonstrator action coverage does not give a guarantee about the sample complexity of the downstream RL finetuning. Can we derive a non-trivial *sufficient* condition that ensures efficient RL finetuning without the aid of exploration approaches typically absent in practice (such as optimism), and how can we pretrain policies to ensure they meet such a sufficient condition?

- We have focused on pretraining only with supervised learning. While this is the most scalable approach, and the most commonly used approach in practice, is this a limiting factor in obtaining an effective initialization for online RL finetuning, and could we pretrain using other approaches as well (for example, offline RL)?

- While we have primarily considered applications to robotic control, our approach could also be applied in language domains. Does pretraining (or SFT finetuning) of language models with our approach lead to improved performance in downstream RL finetuning?

## Acknowledgments

This research was partly supported by RAI, ONR N00014-25-1-2060, and NSF IIS-2150826. The work of CF was partially supported by an NSF CAREER award.

## Impact Statement

This paper presents work whose goal is to advance the field of machine learning. There are many potential societal consequences of our work, none of which we feel must be specifically highlighted here.

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

# A. Additional Related Work

**RL finetuning of pretrained policies.** RL finetuning of pretrained policies is a critical step in both language and robotic domains. In language domains, RL finetuning has proved crucial in aligning LLMs to human values (Ziegler et al., 2019; Ouyang et al., 2022; Bai et al., 2022a; Ramamurthy et al., 2022; Touvron et al., 2023), and enabling reasoning abilities (Shao et al., 2024; Team et al., 2025; Guo et al., 2025a). A host of finetuning algorithms have been developed, both online (Bai et al., 2022b; Bakker et al., 2022; Dumoulin et al., 2023; Lee et al., 2023; Munos et al., 2023; Swamy et al., 2024; Chakraborty et al., 2024; Chang et al., 2024) and offline (Rafailov et al., 2023; Azar et al., 2024; Rosset et al., 2024; Tang et al., 2024; Yin et al., 2024). In robotic control domains, RL finetuning methods include directly modifying the weights of the base pretrained policy (Zhang et al., 2024; Xu et al., 2024; Mark et al., 2024; Ren et al., 2024; Hu et al., 2025; Guo et al., 2025b; Lu et al., 2025; Chen et al., 2025c; Liu et al., 2025), Best-of-$N$ sampling-style approaches that filter the output of the pretrained policy with a learned value function (Chen et al., 2022; Hansen-Estruch et al., 2023; He et al., 2024; Nakamoto et al., 2024; Dong et al., 2025b), "steering" the pretrained policy by altering its sampling process (Wagenmaker et al., 2025a), and learning smaller residual policies to augment the pretrained policy's actions (Ankile et al., 2024b; Yuan et al., 2024; Jülg et al., 2025; Dong et al., 2025a). Our work is tangential to this line of work: rather than improving the finetuning algorithm, we aim to ensure the pretrained policy is amenable to RL finetuning.

**Imitation learning and meta-learning.** While our primary focus is on behavioral cloning (as noted, the workhorse of most modern applications) other approaches to pretraining from demonstrations exist. BC is only one possible instantiation of *imitation learning*; other approaches to imitation learning include inverse RL (Ng et al., 2000; Abbeel & Ng, 2004; Ziebart et al., 2008), methods that aim to learn a policy matching the state distribution of the demonstrator, such as adversarial imitation learning (Ho & Ermon, 2016; Kostrikov et al., 2018; Fu et al., 2017; Kostrikov et al., 2019; Ni et al., 2021; Garg et al., 2021; Xu et al., 2022; Li et al., 2023b; Yue et al., 2024), and robust imitation learning (Chae et al., 2022; Desai et al., 2020; Tangkaratt et al., 2020; Wang et al., 2021; Giammarino et al., 2025). The majority of these works, however, either assume access to additional data sources (e.g. suboptimal trajectories), or require online environment access and are therefore not truly offline pretraining approaches, which is the focus of this work. Furthermore, none of these works explicitly consider the role of pretraining in enabling efficient RL finetuning.

Meta-learning directly aims learn an initialization that can be quickly adapted to a new task. While instantiations of meta-learning for imitation learning exist (Duan et al., 2017; Finn et al., 2017b; James et al., 2018; Dasari & Gupta, 2021; Gao et al., 2023), our setting differs fundamentally from the meta-imitation learning setting. Meta-imitation learning assumes access to demonstration data from *more than one task*, and attempts to learn an initialization that will allow for quickly adapting to demonstrations from a *new* task. In contrast, our goal is to obtain an approach able to learn on a one (or potentially multiple) task(s), and we aim to find an initialization that allows for improvement on the *same* task (or set of tasks), while preserving pretrained performance on this task. Furthermore, rather than learning from new *demonstrations*, as meta-imitation learning does, we aim to learn from (potentially suboptimal) data collected online and that is labeled with rewards. Finally, a primary goal of our work is to ensure that the pretrained policy itself performs well (that is, it has effective 0-shot performance, on par with BC). Meta-imitation learning, in contrast, typically does not consider the 0-shot performance of the pretrained policy, but instead considers the 1-shot performance, i.e. the performance after it has already been updated on the new task. This is a fundamentally different constraint on the pretrained policy than is typically present in the meta-learning setting, but is critical in real-world settings where we want a pretrained policy that behaves well 0-shot.

**Reinforcement learning-based pretraining.** In the RL literature, two lines of work bear some resemblance to ours as well. The *offline-to-online RL* setting aims to train policies with RL on offline datasets that can then be improved with further online interaction (Lee et al., 2022; Ghosh et al., 2022; Kumar et al., 2022; Zhang et al., 2023; Uchendu et al., 2023; Zheng et al., 2023; Ball et al., 2023; Nakamoto et al., 2023), and the *meta-RL* setting aims to meta-learn a policy on some set of tasks which can then be quickly adapted to a new task (Wang et al., 2016; Duan et al., 2016; Finn et al., 2017a; 2018). While similar to our work in that these works also aim to learn behaviors that can be efficiently improved online, the settings differ significantly in that the offline- or meta-pretraining typically requires reward labels (rather than unlabeled demonstrations) and are performed with RL (rather than BC)—in contrast, we study how BC-like pretraining (as noted, the workhorse of most modern applications) can enable efficient online adaptation.

**Posterior sampling and exploration.** Our proposed approach relies on modeling the posterior distribution of the demonstrator's behavior. While this is, to the best of our knowledge, the first example of applying posterior sampling to

BC, posterior methods have a long history in RL, going back to the work of (Thompson, 1933). This works spans applied (Osband et al., 2016a;b; 2018; Zintgraf et al., 2019) and theoretical (Agrawal & Goyal, 2012; Russo & Van Roy, 2014; Russo et al., 2018; Janz et al., 2024; Kveton et al., 2020; Russo, 2019) settings. More generally, our approach can be seen as enabling BC-trained policies to *explore* more effectively. Exploration is a well-studied problem in the RL community (Stadie et al., 2015; Bellemare et al., 2016; Burda et al., 2018; Choi et al., 2018; Ecoffet et al., 2019; Shyam et al., 2019; Lee et al., 2021; Henaff et al., 2022), with several works considering learning exploration strategies from offline data (Hu et al., 2023; Li et al., 2023a; Wilcoxson et al., 2024; Wagenmaker et al., 2025b). These works, however, either consider RL-based pretraining (while we focus on BC) or do not consider the question of online finetuning.

## B. Proofs

### B.1. BC Policy Fails to Cover Demonstrator Actions

*Proof of Proposition 2.* Let $\mathcal{M}^1$ and $\mathcal{M}^2$ denote multi-armed bandits with 3 arms and reward functions $r^1$ and $r^2$:

$$r^1(a_1) = 0, r^1(a_2) = 1, r^1(a_3) = 0$$
$$r^2(a_1) = 0, r^2(a_2) = 0, r^2(a_3) = 1.$$

Let $\pi^\beta(a_1) = 1 - 4\epsilon$, $\pi^\beta(a_2) = 2\epsilon$, $\pi^\beta(a_3) = 2\epsilon$.

By construction of $\widehat{\pi}^{\mathrm{bc}}$, if $T(a_2) = 0$ then we will have $\widehat{\pi}^{\mathrm{bc}}(a_2) = 0$, and if $T(a_3) = 0$ we will have $\widehat{\pi}^{\mathrm{bc}}(a_3) = 0$. By the definition of both $\mathcal{M}^1$ and $\mathcal{M}^2$, we have

$$\mathbb{P}^{\mathcal{M}^i}[T(a_2) = 0, T(a_3) = 0] = (1 - 4\epsilon)^T.$$

As we have assumed that $T \le \frac{1}{20\epsilon}$ and $\epsilon \in (0, 1/8]$, some calculation shows that we can lower bound this as $1/2$. Note that for both $\mathcal{M}^1$ and $\mathcal{M}^2$, we have $\mathcal{J}(\pi^\beta) = 2\epsilon$, while for policies $\widehat{\pi}^{\mathrm{bc}}$ that only play $a_1$, we have $\mathcal{J}(\widehat{\pi}^{\mathrm{bc}}) = 0$. This proves the first part of the result.

For the second part, note that the optimal policy on $\mathcal{M}^1$ plays only $a_2$ and has expected reward of 1, while the optimal policy on $\mathcal{M}^2$ plays only $a_3$ and has expected reward of 1. Let $\widehat{\pi}$ denote an estimate of the optimal policy and $\mathbb{E}^{\mathcal{M}^i, \widehat{\pi}^{\mathrm{bc}}}[\cdot]$ the expectation induced by playing the policy $\widehat{\pi}^{\mathrm{bc}}$ from the first part on instance $\mathcal{M}^i$. Then:

$$\min_{\widehat{\pi}} \max_{i \in \{1,2\}} \mathbb{E}^{\mathcal{M}^i, \widehat{\pi}^{\mathrm{bc}}}[\max_\pi \mathcal{J}^{\mathcal{M}^i}(\pi) - \mathcal{J}^{\mathcal{M}^i}(\widehat{\pi})] = \min_{\widehat{\pi}} \max_{i \in \{1,2\}} \mathbb{E}^{\mathcal{M}^i, \widehat{\pi}^{\mathrm{bc}}}[1 - \widehat{\pi}(a_{1+i})].$$

Note that $1 - \widehat{\pi}(a_2) = \widehat{\pi}(a_1) + \widehat{\pi}(a_3) \ge \widehat{\pi}(a_3)$. Thus we can lower bound the above as

$$\ge \min_{\widehat{\pi}} \max\{\mathbb{E}^{\mathcal{M}^1, \widehat{\pi}^{\mathrm{bc}}}[\widehat{\pi}(a_3)], \mathbb{E}^{\mathcal{M}^2, \widehat{\pi}^{\mathrm{bc}}}[1 - \widehat{\pi}(a_3)]\}$$
$$\ge \min_{\widehat{\pi}} \frac{1}{2} \left( \mathbb{E}^{\mathcal{M}^1, \widehat{\pi}^{\mathrm{bc}}}[\widehat{\pi}(a_3)] + \mathbb{E}^{\mathcal{M}^2, \widehat{\pi}^{\mathrm{bc}}}[1 - \widehat{\pi}(a_3)] \right)$$
$$\ge \frac{1}{2} - \frac{1}{2} \min_{\widehat{\pi}} \left| \mathbb{E}^{\mathcal{M}^1, \widehat{\pi}^{\mathrm{bc}}}[\widehat{\pi}(a_3)] - \mathbb{E}^{\mathcal{M}^2, \widehat{\pi}^{\mathrm{bc}}}[\widehat{\pi}(a_3)] \right|.$$

We can bound

$$\left| \mathbb{E}^{\mathcal{M}^1, \widehat{\pi}^{\mathrm{bc}}}[\widehat{\pi}(a_3)] - \mathbb{E}^{\mathcal{M}^2, \widehat{\pi}^{\mathrm{bc}}}[\widehat{\pi}(a_3)] \right| \le \mathrm{TV}(\mathbb{P}^{\mathcal{M}^1, \widehat{\pi}^{\mathrm{bc}}}, \mathbb{P}^{\mathcal{M}^2, \widehat{\pi}^{\mathrm{bc}}}).$$

Since $\mathcal{M}^1$ and $\mathcal{M}^2$ only differ on $a_2$ and $a_3$, and since $\widehat{\pi}^{\mathrm{bc}}(a_2) = \widehat{\pi}^{\mathrm{bc}}(a_3) = 0$, we have $\mathrm{TV}(\mathbb{P}^{\mathcal{M}^1, \widehat{\pi}^{\mathrm{bc}}}, \mathbb{P}^{\mathcal{M}^2, \widehat{\pi}^{\mathrm{bc}}}) = 0$. Thus, we conclude that

$$\min_{\widehat{\pi}} \max_{i \in \{1,2\}} \mathbb{E}^{\mathcal{M}^i, \widehat{\pi}^{\mathrm{bc}}}[\max_\pi \mathcal{J}^{\mathcal{M}^i}(\pi) - \mathcal{J}^{\mathcal{M}^i}(\widehat{\pi})] \ge \frac{1}{2}.$$

This proves the second part of the result.

$\square$

### B.2. Uniform Noise Fails

*Proof of Proposition 3.* **Construction.** Let $\mathcal{M}$ be the MDP with state space $\{\widetilde{s}_1, \ldots, \widetilde{s}_k, s_1, s_2\}$, actions $\{a_1, a_2\}$, horizon $H \geq 2$ with initial state distribution:

$$P_0(s_1) = 1/2, \quad P_0(\widetilde{s}_1) = 2^{-2} + 2^{-k}, \quad P_0(\widetilde{s}_i) = 2^{-i-1}, i \geq 2,$$

transition function, for all $h \in [H]$:

$$P_h(\widetilde{s}_i \mid \widetilde{s}_i, a) = 1, \forall a \in \mathcal{A}, \quad P_h(s_1 \mid s_1, a_1) = 1,$$
$$P_h(s_2 \mid s_1, a_2) = 1, \quad P_h(s_2 \mid s_2, a) = 1, \forall a \in \mathcal{A},$$

and reward that is 0 everywhere except

$$r_1(\widetilde{s}_i, a_1) = r_H(s_1, a_1) = 1, \quad r_1(\widetilde{s}_i, a_2) = 1 - 2\Delta,$$

for some $\Delta > 0$ to be specified. We consider $\pi^\beta$ defined as

$$\pi_h^\beta(a_1 \mid \widetilde{s}_i) = \pi_h^\beta(a_2 \mid \widetilde{s}_i) = \frac{1}{2}, \quad \pi_h^\beta(a_1 \mid s_1) = 1.$$

Let $\epsilon := \frac{H^2 S \log T}{T} + \xi$, and set $\Delta \leftarrow 2\epsilon$.

**Upper bound on $\alpha$.** Note that $\mathcal{J}(\pi^\beta) = 1 - \frac{1}{2}\Delta$, and that the value of the optimal policy $\pi^\star$ is $\mathcal{J}(\pi^\star) = \max_\pi \mathcal{J}(\pi) = 1$. Let $\widetilde{\pi}^{\mathrm{u},\alpha}$ denote the policy that, on all $\widetilde{s}_i$ plays $\pi^\star$, and on other states plays $\pi^\star$ with probability $1 - \alpha$, and otherwise plays $\mathrm{unif}(\mathcal{A})$. Note then that, regardless of the value of $\widehat{\pi}^{\mathrm{bc}}$, we have that $\mathcal{J}(\widetilde{\pi}^{\mathrm{u},\alpha}) \geq \mathcal{J}(\widehat{\pi}^{\mathrm{u},\alpha})$. Thus,

$$\mathcal{J}(\pi^\beta) - \mathbb{E}[\mathcal{J}(\widehat{\pi}^{\mathrm{u},\alpha})] \geq \mathcal{J}(\pi^\beta) - \mathcal{J}(\widetilde{\pi}^{\mathrm{u},\alpha})$$

If we are in $s_1$ at $h = 2$, the only way we can receive any reward on the episode is if we take action $a_1$ for the last $H - 1$ steps, and we then receive a reward of 1. Under $\widetilde{\pi}^{\mathrm{u},\alpha}$, we take $a_1$ at each step with probability $1 - \alpha + \alpha/A$, so our probability of getting a reward of 1 is $(1 - \alpha + \alpha/A)^{H-1}$. Note that in contrast $\pi^\beta$ will always play $a_1$ and receive a reward of 1 in this situation. If we are in $\widetilde{s}_i$ at $h = 2$ for any $i$, then $\pi^\beta$ will incur a loss of $\Delta$ more than $\widetilde{\pi}^{\mathrm{u},\alpha}$. Thus, we can lower bound

$$\mathcal{J}(\pi^\beta) - \mathcal{J}(\widetilde{\pi}^{\mathrm{u},\alpha}) \geq -\frac{1}{2}\Delta + \frac{1}{2} \cdot (1 - (1 - \alpha + \alpha/A)^{H-1})$$

By assumption we have that $\frac{1}{2}\Delta = \epsilon$. Thus, if we want $\mathcal{J}(\pi^\beta) - \mathbb{E}[\mathcal{J}(\widehat{\pi}^{\mathrm{u},\alpha})] \leq \epsilon$, we need

$$\frac{1}{2} \cdot (1 - (1 - \alpha + \alpha/A)^{H-1}) \leq 2\epsilon.$$

Rearranging this, we have

$$1 - 4\epsilon \leq (1 - \alpha + \alpha/A)^{H-1} \iff \frac{1}{H-1} \log(1 - 4\epsilon) \leq \log(1 - \alpha + \alpha/A).$$

From the Taylor decomposition of $\log(1 - x)$, we see that $\log(1 - \alpha + \alpha/A) \leq -(1 - 1/A)\alpha$. Furthermore, we can lower bound

$$\log(1 - 4\epsilon) \geq -8\epsilon$$

as long as $\epsilon \leq 1/2$. Altogether, then, we have

$$\frac{-8\epsilon}{H-1} \leq -(1 - 1/A)\alpha \implies \alpha \leq \frac{8\epsilon}{(H-1)(1 - 1/A)} \implies \alpha \leq 32\epsilon$$

where the last inequality follows since $H \geq 2, A = 2$.

**Upper bound on $\gamma$.** Let $i_T := \arg\max_i\{2^{-i-1} \mid 2^{-i-1} \le 1/T\}$, so that $1/2T \le P_0(\widetilde{s}_{i_T}) \le 1/T$, and note that such an $\widetilde{s}_{i_T}$ exists by construction. Let $\mathcal{E}$ be the event $\mathcal{E} := \{T_1(\widetilde{s}_{i_T}) = T_1(\widetilde{s}_{i_T}, a_2) = 1\}$. We have

$$
\begin{aligned}
\mathbb{P}[\mathcal{E}] &= \mathbb{P}[T_1(\widetilde{s}_{i_T}, a_2) = 1 \mid T_1(\widetilde{s}_{i_T}) = 1]\mathbb{P}[T_1(\widetilde{s}_{i_T}) = 1] \\
&= \frac{1}{2} \cdot TP_0(\widetilde{s}_{i_T})(1 - P_0(\widetilde{s}_{i_T}))^{T-1} \\
&= \frac{1}{2} \cdot T \cdot \frac{1}{2T} \cdot (1 - \frac{1}{T})^{T-1} \\
&\ge \frac{1}{4e}.
\end{aligned}
$$

Note that on the event $\mathcal{E}$, we have $\widehat{\pi}_1^{\mathrm{bc}}(a_1 \mid \widetilde{s}_{i_T}) = 0$, but $\pi_1^\beta(a_1 \mid \widetilde{s}_{i_T}) = 1/2$. Thus,

$$
\widehat{\pi}_1^{\mathrm{u},\alpha}(a_1 \mid \widetilde{s}_{i_T}) = \alpha/A \le 32\epsilon/A = 64\epsilon/A \cdot \pi_1^\beta(a_1 \mid \widetilde{s}_{i_T})
$$

where we have used the bound on $\alpha$ shown above. Thus, on $\mathcal{E}$, we will only have that $\widehat{\pi}^{\mathrm{u},\alpha}$ achieves demonstrator action coverage for $\gamma \le 64\epsilon/A$. Since $\mathcal{E}$ occurs with probability at least $1/4e$, it follows that if we want to guarantee $\widehat{\pi}^{\mathrm{u},\alpha}$ achieves demonstrator action coverage with probability at least $1 - \delta$ for $\delta < 1/4e$, we must have $\gamma \le 64\epsilon/A$.

Note as well that, since $\widehat{\pi}_1^{\mathrm{bc}}(a_2 \mid \widetilde{s}_{i_T}) = 1$, any policy in the support of $\widehat{\pi}^{\mathrm{bc}}$ will be suboptimal by a factor of at least $P_0(\widetilde{s}_{i_T}) \cdot 2\Delta \ge \Delta/T$. $\qquad\square$

### B.3. Analysis of Posterior Demonstrator Policy

In the tabular setting, some algebra shows that

$$
\widehat{\pi}_h^{\mathrm{post}}(a \mid s) = \begin{cases} \frac{T_h(s,a)+1}{T_h(s)+A} & T_h(s) > 0 \\ \mathrm{unif}(\mathcal{A}) & T_h(s) = 0. \end{cases}
$$

Throughout this section we denote

$$
\widetilde{\pi}_h(a \mid s) := \begin{cases} (1-\alpha) \cdot \frac{T_h(s,a)}{T_h(s)} + \alpha \cdot \frac{T_h(s,a)+\lambda/A}{T_h(s)+\lambda} & T_h(s) > 0 \\ \mathrm{unif}(\mathcal{A}) & T_h(s) = 0 \end{cases}
$$

for some $\alpha \in [0,1]$.

We also denote $w_h^\pi(s,a) := \mathbb{P}^\pi[s_h = s, a_h = a]$. $Q_h^\pi(s,a) := \mathbb{E}^\pi[\sum_{h' \ge h} r_{h'}(s_{h'}, a_{h'}) \mid s_h = s, a_h = a]$ denotes the standard $Q$-function. $\mathcal{J}(\pi; r)$ denotes the expected return of policy $\pi$ for reward $r$.

**Lemma 1.** *As long as $\delta \le 0.9$ and $\lambda \ge A$, we have*

$$
\mathbb{P}\left[\widetilde{\pi}_h(a \mid s) \ge \alpha \cdot \min\left\{\frac{\pi_h^\beta(a \mid s)}{64\log SH/\delta}, \frac{1}{2\lambda}\right\}, \forall a \in \mathcal{A}, s \in \mathcal{S}, h \in [H]\right] \ge 1 - \delta.
$$

*Proof.* Consider some $(s,h)$. By Bernstein's inequality, if $T_h(s) > 0$, we have that with probability at least $1 - \delta$,

$$
\frac{T_h(s,a)}{T_h(s)} \ge \pi_h^\beta(a \mid s) - \sqrt{\frac{2\pi_h^\beta(a \mid s)\log 1/\delta}{T_h(s)}} - \frac{2\log 1/\delta}{3T_h(s)}. \tag{3}
$$

From some algebra, we see that as long as $T_h(s) \ge \frac{32\log 1/\delta}{\pi_h^\beta(a|s)}$, we have that $\frac{T_h(s,a)}{T_h(s)} \ge \frac{1}{2}\pi_h^\beta(a \mid s)$. By the definition of $\widetilde{\pi}$,

under the good event of (3) we can then lower bound

$$
\widetilde{\pi}_h(a \mid s) \geq \begin{cases} \frac{\alpha}{1+\lambda/T_h(s)} \cdot \frac{1}{2}\pi_h^\beta(a \mid s) & T_h(s) \geq \frac{32\log 1/\delta}{\pi_h^\beta(a|s)} \\ \frac{\alpha\lambda/A}{T_h(s)+A} & \text{o.w.} \end{cases}
$$

$$
\geq \begin{cases} \frac{\alpha \cdot 32\log 1/\delta}{32\log 1/\delta + \lambda \cdot \pi_h^\beta(a|s)} \cdot \frac{1}{2}\pi_h^\beta(a \mid s) & N_h(s) \geq \frac{32\log 1/\delta}{\pi_h^\beta(a|s)} \\ \frac{\alpha\lambda/A \cdot \pi_h^\beta(a|s)}{32\log 1/\delta + \lambda \cdot \pi_h^\beta(a|s)} & \text{o.w.} \end{cases}
$$

$$
\overset{(a)}{\geq} \frac{\alpha \cdot \pi_h^\beta(a \mid s)}{32\log 1/\delta + \lambda \cdot \pi_h^\beta(a \mid s)}
$$

$$
\geq \alpha \cdot \min\left\{ \frac{\pi_h^\beta(a \mid s)}{64\log 1/\delta}, \frac{1}{2\lambda} \right\}
$$

where $(a)$ follows as long as $\delta \leq 0.9$ and $\lambda \geq A$. In the case when $T_h(s) = 0$ we have $\widetilde{\pi}_h(a \mid s) = 1/A \geq 1/\lambda$, so this lower bound still holds. Taking a union bound over arms proves the result. $\qquad\square$

**Lemma 2.** *As long as $\lambda \geq 4\log(HT)$, we have*

$$
\mathbb{E}[\mathcal{J}(\widehat{\pi}^{\mathrm{bc}}) - \mathcal{J}(\widetilde{\pi})] \lesssim (1+\alpha H) \cdot \frac{H^2 S \log T}{T} + \alpha \cdot \frac{H^2 S \lambda}{T}.
$$

*Proof.* By the Performance-Difference Lemma we have:

$$
\mathcal{J}(\widehat{\pi}^{\mathrm{bc}}) - \mathcal{J}(\widetilde{\pi}) = \sum_{h=1}^{H}\sum_{s\in\mathcal{S}} w_h^{\widehat{\pi}^{\mathrm{bc}}}(s) \cdot \left( \mathbb{E}_{a\sim\widehat{\pi}_h^{\mathrm{bc}}(s)}[Q_h^{\widetilde{\pi}}(s,a)] - \mathbb{E}_{a\sim\widetilde{\pi}_h(s)}[Q_h^{\widetilde{\pi}}(s,a)] \right)
$$

$$
\leq \sum_{h=1}^{H}\sum_{s\in\mathcal{S}} w_h^{\widehat{\pi}^{\mathrm{bc}}}(s) \cdot \left| \mathbb{E}_{a\sim\widehat{\pi}_h^{\mathrm{bc}}(s)}[Q_h^{\widetilde{\pi}}(s,a)] - \mathbb{E}_{a\sim\widetilde{\pi}_h(s)}[Q_h^{\widetilde{\pi}}(s,a)] \right|. \tag{4}
$$

For $(s,h)$ with $N_h(s) > 0$, we have

$$
\left| \mathbb{E}_{a\sim\widehat{\pi}_h^{\mathrm{bc}}(s)}[Q_h^{\widetilde{\pi}}(s,a)] - \mathbb{E}_{a\sim\widetilde{\pi}_h(s)}[Q_h^{\widetilde{\pi}}(s,a)] \right| \leq \sum_{a\in\mathcal{A}} H \cdot |\widehat{\pi}_h^{\mathrm{bc}}(a \mid s) - \widetilde{\pi}_h(a \mid s)|,
$$

where we have used that $Q_h^{\widehat{\pi}^{\mathrm{post}}}(s,a) \in [0, H]$. Then, using the definition of $\widehat{\pi}^{\mathrm{bc}}$ and $\widetilde{\pi}$ we can bound this as

$$
\leq \sum_{a\in\mathcal{A}} \alpha H \cdot \left| \frac{T_h(s,a)}{T_h(s)} - \frac{T_h(s,a) + \lambda/A}{T_h(s) + \lambda} \right|
$$

$$
= \sum_{a\in\mathcal{A}} \frac{\alpha\lambda H}{A} \cdot \left| \frac{AT_h(s,a) - T_h(s)}{T_h(s)(T_h(s) + \lambda)} \right|
$$

$$
\leq \sum_{a\in\mathcal{A}} \frac{\alpha\lambda H}{A} \cdot \frac{AT_h(s,a) + T_h(s)}{T_h(s)(T_h(s) + \lambda)}
$$

$$
= \frac{2\alpha\lambda H}{T_h(s) + \lambda}.
$$

Since $\mathbb{E}_{a\sim\widehat{\pi}_h^{\mathrm{bc}}(s)}[Q_h^{\widetilde{\pi}}(s,a)] - \mathbb{E}_{a\sim\widetilde{\pi}_h(s)}[Q_h^{\widetilde{\pi}}(s,a)] = 0$ by construction when $T_h(s) = 0$, we then have

$$
(4) \leq \sum_{h=1}^{H}\sum_{s\in\mathcal{S}} w_h^{\widehat{\pi}^{\mathrm{bc}}}(s) \cdot \frac{2\alpha\lambda H}{T_h(s) + \lambda}.
$$

Let $\mathcal{E}$ denote the good event from Lemma 3 with $\delta = \frac{S}{T}$. Then as long as $\lambda \geq 4\log(HT)$ we can bound the above as

$$\leq \sum_{h=1}^{H}\sum_{s\in\mathcal{S}} w_h^{\widehat{\pi}^{\mathrm{bc}}}(s) \cdot \frac{2\alpha\lambda H}{T_h(s) + \lambda}\mathbb{I}\{\mathcal{E}\} + 2H^2 \cdot \mathbb{I}\{\mathcal{E}^c\}$$

$$\leq \sum_{h=1}^{H}\sum_{s\in\mathcal{S}} w_h^{\widehat{\pi}^{\mathrm{bc}}}(s) \cdot \frac{4\alpha\lambda H}{w_h^{\pi^\beta}(s) \cdot T + \lambda} + 2H^2 \cdot \mathbb{I}\{\mathcal{E}^c\}.$$

Let $\widetilde{r}$ denote the reward function:

$$\widetilde{r}_h(s,a) := \frac{\lambda}{w_h^{\pi^\beta}(s) \cdot T + \lambda}$$

and note that $\widetilde{r} \in [0,1]$, and

$$\sum_{h=1}^{H}\sum_{s\in\mathcal{S}} w_h^{\widehat{\pi}^{\mathrm{bc}}}(s) \cdot \frac{4\alpha\lambda H}{w_h^{\pi^\beta}(s) \cdot T + \lambda} = 4\alpha H \cdot \mathcal{J}(\widehat{\pi}^{\mathrm{bc}}; \widetilde{r}).$$

By Theorem 4.4 of (Rajaraman et al., 2020), we have[1]

$$\mathbb{E}[\mathcal{J}(\widehat{\pi}^{\mathrm{bc}}; \widetilde{r})] \lesssim \mathcal{J}(\pi^\beta; \widetilde{r}) + \frac{H^2 S \log T}{T}$$

$$= \sum_{h=1}^{H}\sum_{s\in\mathcal{S}} w_h^{\pi^\beta}(s) \cdot \frac{\lambda}{w_h^{\pi^\beta}(s) \cdot T + \lambda} + \frac{H^2 S \log T}{T}$$

$$\leq \frac{HS\lambda}{T} + \frac{H^2 S \log T}{T}.$$

Noting that $\mathbb{E}[2H^2 \cdot \mathbb{I}\{\mathcal{E}^c\}] \leq 2H^2\delta \leq \frac{2H^2 S}{T}$ completes the proof. $\qquad\square$

**Lemma 3.** *With probability at least $1 - \delta$, for all $(s,h)$, we have*

$$T_h(s) + \lambda \geq \frac{1}{2}w_h^{\pi^\beta}(s) \cdot T + \frac{1}{2}\lambda$$

*as long as $\lambda \geq 4\log\frac{SH}{\delta}$.*

*Proof.* Consider some $(s,h)$ and note that $\mathbb{E}[T_h(s)/T] = w_h^{\pi^\beta}(s)$. By Bernstein's inequality, we have with probability $1 - \delta/SH$:

$$T_h(s) \geq w_h^{\pi^\beta}(s) \cdot T - \sqrt{2w_h^{\pi^\beta}(s) \cdot T \cdot \log\frac{SH}{\delta}} - \frac{2}{3}\log\frac{SH}{\delta}.$$

We would then like to show that

$$w_h^{\pi^\beta}(s) \cdot T - \sqrt{2w_h^{\pi^\beta}(s) \cdot T \cdot \log\frac{SH}{\delta}} - \frac{2}{3}\log\frac{SH}{\delta} + \lambda \geq \frac{1}{2}(w_h^{\pi^\beta}(s) \cdot T + \lambda)$$

$$\iff \frac{1}{2}w_h^{\pi^\beta}(s) \cdot T + \frac{1}{2}\lambda \geq \sqrt{2w_h^{\pi^\beta}(s) \cdot T \cdot \log\frac{SH}{\delta}} + \frac{2}{3}\log\frac{SH}{\delta}$$

As we have assumed $\lambda \geq 4\log\frac{SH}{\delta}$, it suffices to show

$$\frac{1}{2}w_h^{\pi^\beta}(s) \cdot T + \log\frac{SH}{\delta} \geq \sqrt{2w_h^{\pi^\beta}(s) \cdot T \cdot \log\frac{SH}{\delta}}.$$

However, this is true by the AM-GM inequality. A union bound proves the result. $\qquad\square$

---

[1]Note that Theorem 4.4 of (Rajaraman et al., 2020) shows an inequality in the opposite direction of what we show here: they bound $\mathcal{J}(\pi^\beta; \widetilde{r}) - \mathbb{E}[\mathcal{J}(\widehat{\pi}^{\mathrm{bc}}; \widetilde{r})]$ instead of $\mathbb{E}[\mathcal{J}(\widehat{\pi}^{\mathrm{bc}}; \widetilde{r})] - \mathcal{J}(\pi^\beta; \widetilde{r})$. However, we see that the only place in their proof where their argument relied on this ordering is in Lemma A.8. We show in Lemma 4 that a reverse version of their Lemma A.8 holds, allowing us to instead bound $\mathbb{E}[\mathcal{J}(\widehat{\pi}^{\mathrm{bc}}; \widetilde{r})] - \mathcal{J}(\pi^\beta; \widetilde{r})$.

**Lemma 4** (Reversed version of Lemma A.8 of (Rajaraman et al., 2020)). *Adopting the notation from (Rajaraman et al., 2020), we have*

$$\mathbb{E}[\Pr_{\pi^{\mathrm{first}}}[\mathcal{E}]] \leq \frac{SH \log N}{N}$$

*for $\mathcal{E}^c$ the event that within a trajectory, the policy only visits states for which $T_h(s) > 0$.*

*Proof.* Let $\mathcal{E}_{s,h}$ denote the event that the state $s$ is visited at step $h$ and $T_h(s) = 0$, and $\mathcal{E}_h := \cup_{s \in \mathcal{S}} \mathcal{E}_{s,h}$. Then, by simple set inclusions, we have:

$$\mathcal{E} = \bigcup_{h \in [H]} \bigcup_{s \in \mathcal{S}} \mathcal{E}_{s,h} = \bigcup_{h \in [H]} \bigcup_{s \in \mathcal{S}} \left( \mathcal{E}_{s,h} \cap \bigcap_{h' < h} \mathcal{E}_{h'}^c \right).$$

By a union bound it follows that

$$\mathbb{E}[\Pr_{\pi^{\mathrm{first}}}[\mathcal{E}]] \leq \sum_{h \in [H]} \sum_{s \in \mathcal{S}} \mathbb{E}[\Pr_{\pi^{\mathrm{first}}}[\mathcal{E}_{s,h} \cap \bigcap_{h' < h} \mathcal{E}_{h'}^c]].$$

Now note that

$$\Pr_{\pi^{\mathrm{first}}}[\mathcal{E}_{s,h} \cap \bigcap_{h' < h} \mathcal{E}_{h'}^c] = \Pr_{\pi^{\mathrm{first}}}[\mathcal{E}_{s,h} \mid \bigcap_{h' < h} \mathcal{E}_{h'}^c] \Pr_{\pi^{\mathrm{first}}}[\bigcap_{h' < h} \mathcal{E}_{h'}^c]$$

$$= \Pr_{\pi^{\mathrm{first}}}[\mathcal{E}_{s,h} \mid \bigcap_{h' < h} \mathcal{E}_{h'}^c] \Pr_{\pi^{\mathrm{first}}}[\mathcal{E}_{h-1}^c \mid \bigcap_{h' < h-1} \mathcal{E}_{h'}^c] \Pr_{\pi^{\mathrm{first}}}[\bigcap_{h' < h-1} \mathcal{E}_{h'}^c]$$

$$\vdots$$

$$= \Pr_{\pi^{\mathrm{first}}}[\mathcal{E}_{s,h} \mid \bigcap_{h' < h} \mathcal{E}_{h'}^c] \cdot \prod_{h' < h} \Pr_{\pi^{\mathrm{first}}}[\mathcal{E}_{h'}^c \mid \bigcap_{h'' < h'} \mathcal{E}_{h''}^c].$$

If the event $\bigcap_{h' < h} \mathcal{E}_{h'}^c$ holds, then up to step $h$ no states are encountered for which $T_{h'}(s) = 0$. Thus, on such states, $\pi^{\mathrm{first}}$ and $\pi^{\mathrm{orc-first}}$ will behave identically. It follows that $\mathbb{E}[\Pr_{\pi^{\mathrm{first}}}[\mathcal{E}_{s,h} \mid \bigcap_{h' < h} \mathcal{E}_{h'}^c]] = \mathbb{E}[\Pr_{\pi^{\mathrm{orc-first}}}[\mathcal{E}_{s,h} \mid \bigcap_{h' < h} \mathcal{E}_{h'}^c]]$. By a similar argument, we have $\Pr_{\pi^{\mathrm{orc-first}}}[\mathcal{E}_{h'}^c \mid \bigcap_{h'' < h'} \mathcal{E}_{h''}^c] = \Pr_{\pi^{\mathrm{first}}}[\mathcal{E}_{h'}^c \mid \bigcap_{h'' < h'} \mathcal{E}_{h''}^c]$ for each $h' < h$. Thus,

$$\Pr_{\pi^{\mathrm{first}}}[\mathcal{E}_{s,h} \cap \bigcap_{h' < h} \mathcal{E}_{h'}^c] = \Pr_{\pi^{\mathrm{orc-first}}}[\mathcal{E}_{s,h} \cap \bigcap_{h' < h} \mathcal{E}_{h'}^c].$$

It follows that

$$\mathbb{E}[\Pr_{\pi^{\mathrm{first}}}[\mathcal{E}]] \leq \sum_{h \in [H]} \sum_{s \in \mathcal{S}} \mathbb{E}[\Pr_{\pi^{\mathrm{orc-first}}}[\mathcal{E}_{s,h} \cap \bigcap_{h' < h} \mathcal{E}_{h'}^c]] \leq \sum_{h \in [H]} \sum_{s \in \mathcal{S}} \mathbb{E}[\Pr_{\pi^{\mathrm{orc-first}}}[\mathcal{E}_{s,h}]].$$

From here the proof follows identically to the proof of Lemma A.8 of (Rajaraman et al., 2020). □

*Proof of Theorem 1.* Set $\lambda = \max\{A, 4\log(HT)\}$ and $\alpha = \frac{1}{\max\{A, H, \log(HT)\}}$. We have

$$\mathcal{J}(\pi^\beta) - \mathbb{E}[\mathcal{J}(\widehat{\pi}^{\mathrm{bc}})] + \mathbb{E}[\mathcal{J}(\widehat{\pi}^{\mathrm{bc}})] - \mathbb{E}[\mathcal{J}(\widetilde{\pi})] \lesssim \frac{H^2 S \log T}{T} + (1 + \alpha H) \cdot \frac{H^2 S \log T}{T} + \alpha \cdot \frac{H^2 S \lambda}{T}$$

where we bound $\mathcal{J}(\pi^\beta) - \mathbb{E}[\mathcal{J}(\widehat{\pi}^{\mathrm{bc}})]$ by Theorem 4.4 of (Rajaraman et al., 2020), and $\mathbb{E}[\mathcal{J}(\widehat{\pi}^{\mathrm{bc}})] - \mathbb{E}[\mathcal{J}(\widetilde{\pi})]$ by Lemma 2 since $\lambda \geq 4\log(HT)$. By our choice of $\alpha = \frac{1}{\max\{A, H, \log(HT)\}}$, we can bound all of this as

$$\lesssim \frac{H^2 S \log T}{T}.$$

This proves the suboptimality guarantee. To show that $\widetilde{\pi}$ achieves demonstrator action coverage, we apply Lemma 1 using our values of $\lambda$ and $\alpha$. □

## B.4. Optimality of Posterior Demonstrator Policy

Let $\mathcal{M}$ denote a multi-armed bandit with $A > 1$ actions where $r(a_1) = 1$ and $r(a_i) = 0$ for $i > 1$. Let $\pi^{\beta,i}$ denote the policy defined as

$$\pi^{\beta,i}(a) = \begin{cases} 1 - \alpha & a = 1 \\ \alpha & a = i \\ 0 & \text{o.w.} \end{cases}$$

for $i > 1$ and $\alpha$ some value we will set, and $\pi^{\beta,1}(1) = 1$. We let $\mathcal{M}^i = (\mathcal{M}, \pi^{\beta,i})$ the instance-demonstrator pair, $\mathbb{E}^i[\cdot]$ the expectation on this instance, $\mathbb{P}^i$ the distribution on this instance, and $\mathbb{P}^{i,T} = \otimes_{t=1}^T \mathbb{P}^i$.

**Lemma 5.** *Consider the instance constructed above. Then we have that, for $j \neq i$:*

$$\mathbb{P}^i[\widehat{\pi}(i) \geq \gamma \cdot \alpha] \leq 2 \cdot \mathbb{P}^j[\widehat{\pi}(i) \geq \gamma \cdot \alpha] + T \cdot \alpha.$$

*Proof.* This follows from Lemma A.11 of (Foster et al., 2021), which immediately gives that:

$$\mathbb{P}^i[\{\widehat{\pi}(i) \geq \gamma \cdot \alpha] \leq 2 \cdot \mathbb{P}^j[\widehat{\pi}(i) \geq \gamma \cdot \alpha] + D_{\mathrm{H}}^2(\mathbb{P}^{i,T}, \mathbb{P}^{j,T}),$$

where $D_{\mathrm{H}}(\cdot, \cdot)$ denotes the Hellinger distance. Since the squared Hellinger distance is subadditive we have

$$D_{\mathrm{H}}^2(\mathbb{P}^{i,T}, \mathbb{P}^{j,T}) \leq T \cdot D_{\mathrm{H}}^2(\mathbb{P}^i, \mathbb{P}^j).$$

By elementary calculations we see that $D_{\mathrm{H}}^2(\mathbb{P}^i, \mathbb{P}^j) = \alpha$, which proves the result. $\square$

**Theorem 3** (Full version of Theorem 2). *Let $\widehat{\pi}$ achieve demonstrator action coverage with some parameter $\gamma$ for each $\mathcal{M}^i, i \in [A]$, and some $\delta \in (0, 1/4]$, and assume that*

$$\mathcal{J}(\pi^{\beta,i}) - \mathbb{E}^i[\mathcal{J}(\widehat{\pi})] \leq \xi, \quad \forall i \geq 1$$

*for some $\xi > 0$. Then if $T \leq \frac{1}{4\alpha}$, it must be the case that*

$$\gamma \leq \frac{\xi}{2A\alpha}.$$

*In particular, setting $\xi = c \cdot \frac{\log T}{T}$ and if $\alpha = \frac{1}{2T}$, we have*

$$\gamma \leq c \cdot \frac{\log T}{A}.$$

*Proof.* Our goal is to find the maximum value of $\gamma$ such that our constraint on the optimality of $\widehat{\pi}$ is met, for each $\mathcal{M}^i$. In particular, this can be upper bounded as

$$\max_{\widehat{\pi}, \gamma} \gamma \quad \text{s.t.} \quad \mathbb{P}^i[\{\widehat{\pi}(a) \geq \gamma \cdot \pi^\beta(a), \forall a \in \mathcal{A}\}] \geq 1 - \delta, \ \mathcal{J}(\pi^{\beta,i}) - \mathbb{E}^i[\mathcal{J}(\widehat{\pi})] \leq \xi, \ \forall i \geq 1. \tag{5}$$

Note that for $\mathcal{M}^i, i \geq 1$, the event $\{\widehat{\pi}(a) \geq \gamma \cdot \pi^{\beta,i}(a), \forall a \in \mathcal{A}\}$ is a subset of the event $\{\widehat{\pi}(i) \geq \gamma \cdot \alpha\}$. This allows us to bound (5) as

$$\max_{\widehat{\pi}, \gamma} \gamma \quad \text{s.t.} \quad \mathbb{P}^i[\widehat{\pi}(i) \geq \gamma \cdot \alpha] \geq 1 - \delta, \ \mathcal{J}(\pi^{\beta,i}) - \mathbb{E}^i[\mathcal{J}(\widehat{\pi})] \leq \xi, \ \forall i \geq 1. \tag{6}$$

By Lemma 5, we have that for each $i > 1$,

$$\mathbb{P}^i[\widehat{\pi}(i) \geq \gamma \cdot \alpha] \leq 2 \cdot \mathbb{P}^1[\widehat{\pi}(i) \geq \gamma \cdot \alpha] + T \cdot \alpha.$$

Furthermore, on $\mathcal{M}^1$ we have $\mathcal{J}(\pi^{\beta,1}) - \mathbb{E}^1[\mathcal{J}(\widehat{\pi})] = \mathbb{E}^1[\sum_{i>1} \widehat{\pi}(i)]$. Given this, we can upper bound (6) as

$$\max_{\widehat{\pi}, \gamma} \gamma \quad \text{s.t.} \quad \mathbb{P}^1[\widehat{\pi}(i) \geq \gamma \cdot \alpha] \geq \frac{1}{2} \cdot (1 - \delta - T \cdot \alpha), \forall i > 1, \ \mathbb{E}^1[\sum_{i>1} \widehat{\pi}(i)] \leq \xi. \tag{7}$$

By Markov's inequality, we have

$$\mathbb{P}^1[\widehat{\pi}(i) \geq \gamma \cdot \alpha] \leq \frac{\mathbb{E}^1[\widehat{\pi}(i)]}{\gamma \cdot \alpha}.$$

Furthermore, since we have assumed $\delta \leq 1/4$ and $T \leq \frac{1}{4\alpha}$, we have $\frac{1}{2} \cdot (1 - \delta - T \cdot \alpha) \geq \frac{1}{4}$. We can therefore bound (7) as

$$\max_{\widehat{\pi}, \gamma} \gamma \quad \text{s.t.} \quad \mathbb{E}^1[\widehat{\pi}(i)] \geq \frac{1}{4} \cdot \gamma\alpha, \forall i > 1, \ \mathbb{E}^1[\sum_{i>1} \widehat{\pi}(i)] \leq \xi. \tag{8}$$

However, we see then that we immediately have

$$\gamma \leq \frac{\xi}{4(A-1)\alpha}.$$

This proves the result as long as $A > 1$. $\qquad \square$

## C. Posterior Demonstrator Policy for Gaussian Demonstrator

Let $P(\cdot \mid \mu)$ denote the distribution $\mathcal{N}(\mu, \Sigma)$, where we assume $\mu$ is unknown and $\Sigma$ is known. Assume that we have samples $\mathfrak{D} = \{x_1, \ldots, x_T\} \sim P(\cdot \mid \mu^\star)$. Let $Q_{\text{prior}} = \mathcal{N}(0, \Lambda_0)$ denote the prior on $\mu$. Throughout this section we let $=^d$ denote equality in distribution.

**Lemma 6.** *Under $Q_{\text{prior}}$, we have that the posterior $Q_{\text{post}}$ on $\mu$ is:*

$$Q_{\text{post}}(\cdot \mid \mathfrak{D}) = \mathcal{N}\left(\Lambda_{\text{post}} \Sigma^{-1} \cdot \sum_{t=1}^{T} x_t, \Lambda_{\text{post}}\right),$$

*for $\Lambda_{\text{post}}^{-1} = \Lambda_0^{-1} + T \cdot \Sigma^{-1}$.*

*Proof.* Dropping terms that do not depend on $\mu$, we have

$$Q_{\text{post}}(\mu \mid \mathfrak{D}) = \frac{P(\mathfrak{D} \mid \mu) Q_{\text{prior}}(\mu)}{P(\mathfrak{D})}$$

$$\propto \exp\left(-\frac{1}{2} \sum_{t=1}^{T} (x_t - \mu)^\top \Sigma^{-1} (x_t - \mu)\right) \cdot \exp\left(-\frac{1}{2} \mu^\top \Lambda_0 \mu\right)$$

$$\propto \exp\left(-\frac{1}{2} T \mu^\top \Sigma^{-1} \mu - \frac{1}{2} \mu^\top Q_{\text{prior}}^{-1} \mu + \mu^\top \Sigma^{-1} \cdot \sum_{t=1}^{T} x_t\right)$$

$$= \exp\left(-\frac{1}{2} (\mu - \Lambda_{\text{post}} v)^\top \Lambda_{\text{post}}^{-1} (\mu - \Lambda_{\text{post}} v) + \frac{1}{2} v^\top \Lambda_{\text{post}} v\right)$$

for $\Lambda_{\text{post}}^{-1} = \Lambda_0^{-1} + T \cdot \Sigma^{-1}$, and $v = \Sigma^{-1} \cdot \sum_{t=1}^{T} x_t$. $\qquad \square$

**Lemma 7** (General version of Proposition 4)**.** *Let*

$$\widehat{\mu} = \arg\min_{\mu} \sum_{t=1}^{T} (\mu - \widetilde{x}_t)^\top \Sigma^{-1} (\mu - \widetilde{x}_t) + (\mu - \widetilde{\mu})^\top \Lambda_0^{-1} (\mu - \widetilde{\mu}),$$

*for $\widetilde{x}_t = x_t + w_t$, $w_t \sim \mathcal{N}(0, \Sigma)$, and $\widetilde{\mu} \sim Q_{\text{prior}}$. Then $\widehat{\mu} =^d Q_{\text{post}}(\cdot \mid \mathfrak{D})$.*

*Proof.* By computing the gradient of the objective, setting it equal to 0, and solving for $\mu$, we see that

$$\widehat{\mu} = (\Lambda_0^{-1} + T\Sigma^{-1})^{-1} \cdot \left(\Sigma^{-1} \cdot \sum_{t=1}^{T} \widetilde{x}_t + \Lambda_0^{-1} \widetilde{\mu}\right)$$

$$= (\Lambda_0^{-1} + T\Sigma^{-1})^{-1} \cdot \Sigma^{-1} \cdot \sum_{t=1}^{T} x_t + (\Lambda_0^{-1} + T\Sigma^{-1})^{-1} \cdot \left(\Sigma^{-1} \cdot \sum_{t=1}^{T} w_t + \Lambda_0^{-1} \widetilde{\mu}\right).$$

| | Pretrained Performance | | | |
| Task | BC | $\sigma$-BC | DICE | POSTBC |
|---|---|---|---|---|
| Robomimic Lift | **71.0** $_{\pm0.5}$ | 68.0 $_{\pm0.7}$ | 25.5 $_{\pm4.4}$ | 69.7 $_{\pm1.5}$ |
| Robomimic Can | 43.1 $_{\pm0.9}$ | 42.3 $_{\pm0.8}$ | 14.2 $_{\pm2.5}$ | **44.7** $_{\pm1.0}$ |
| Robomimc Square | 17.9 $_{\pm0.7}$ | 17.8 $_{\pm0.7}$ | 5.7 $_{\pm0.3}$ | **18.1** $_{\pm0.8}$ |
| Libero Scene 1 (5 tasks) | 23.6 $_{\pm1.7}$ | 22.3 $_{\pm2.1}$ | - | **25.3** $_{\pm1.2}$ |
| Libero Scene 2 (7 tasks) | 11.4 $_{\pm0.2}$ | 10.6 $_{\pm0.8}$ | - | **12.3** $_{\pm1.5}$ |
| Libero Scene 3 (4 tasks) | 39.5 $_{\pm1.5}$ | 36.9 $_{\pm1.8}$ | - | **40.8** $_{\pm1.3}$ |
| Libero All (16 tasks) | 22.2 $_{\pm0.3}$ | 20.9 $_{\pm0.6}$ | - | **23.5** $_{\pm0.5}$ |

*Table 5.* Comparison of success rates of all pretrained policies on `Robomimic` and `Libero`, for different pretraining approaches. Bolded text denotes best approach.

Note that the first term in the above is deterministic conditioned on $\mathfrak{D}$, and the second term is mean 0 and has covariance $(\Lambda_0^{-1} + T\Sigma^{-1})^{-1}$. We see then that the mean and covariance of $\widehat{\mu}$ match the mean the covariance of $Q_{\text{post}}(\cdot \mid \mathfrak{D})$ given in Lemma 6, which proves the result. $\qquad\square$

## D. Additional Experimental Details

We state a more detailed version of our procedure to compute the ensemble for POSTBC in Algorithm 2.

---

**Algorithm 2** Posterior Variance Approximation via Ensembled Prediction

1: **input:** demonstration dataset $\mathfrak{D}$, ensemble size $K$, posterior model class $\mathcal{F}$
2: **for** $\ell = 1, 2, \ldots, K$ **do**
3:     Set $\mathfrak{D}_\ell$ to "noisy" version of $\mathfrak{D}$
4:     Fit $f_\ell$ by solving $f_\ell \leftarrow \arg\min_{f \in \mathcal{F}} \sum_{(s,\widetilde{a}) \in \mathfrak{D}_\ell} \|f_\ell(s) - \widetilde{a}\|_2^2$
   $\mathsf{cov}(\cdot) \leftarrow \sum_{\ell=1}^K (f_\ell(\cdot) - \bar{f}(\cdot))(f_\ell(\cdot) - \bar{f}(\cdot))^\top$ for $\bar{f}(\cdot) \leftarrow \frac{1}{K} \sum_{\ell=1}^K f_\ell(\cdot)$
5: **return** $\mathsf{cov}(\cdot)$

---

Algorithm 2 fits an ensemble of predictors to a perturbed version of $\mathfrak{D}$ in order to approximate a posterior sample, and uses these samples to approximate the posterior covariance. While these samples may not correspond precisely to the posterior covariance in general settings, Proposition 4 shows that in simple settings they do, suggesting that, at minimum, this is a principled approximation. In the Gaussian setting we generate a noisy $\mathfrak{D}$ by perturbing the actions in $\mathfrak{D}$ with Gaussian noise. In practice, however, other methods to obtain a "noisy" version of $\mathfrak{D}$ can be applied as well. In particular, we found that generating $\mathfrak{D}_\ell$ by *bootstrapped sampling* (Fushiki et al., 2005; Osband & Van Roy, 2015; Osband et al., 2016a)—where we sample with replacement from $\mathfrak{D}$—typically outperforms directly adding noise to the actions in $\mathfrak{D}$.

Given the approximate posterior covariance $\mathsf{cov}(\cdot)$, Section 5.1 suggests that for any $s$ we can generate approximate samples from $\widehat{\pi}^{\text{post}}(\cdot \mid s)$ by first sampling an action from the BC policy at $s$, and then perturbing the resulting action by posterior noise $w \sim \mathcal{N}(0, \mathsf{cov}(s))$. In practice, to avoid this two-stage procedure, we can fit a single policy to $\mathfrak{D}$ (which would be the BC policy) but where we perturb each action in $\mathfrak{D}$ by the posterior noise. Note that the distribution this policy will fit is equivalent to the distribution produced by the above two-stage procedure, as long as we utilize an expressive generative model able to represent the posterior demonstrator policy. This is instantiated in Algorithm 1.

For all experiments we instantiate POSTBC directly as suggested in Algorithm 2 and Algorithm 1. We describe additional details on this instantiation next.

In all experiments, we parameterize $f_\ell$ in Algorithm 2 as an MLP (perhaps on top of a ResNet or other feature encoder, as described below). For the `Robomimic` experiments we let $f_\ell$ parameterize a Gaussian distribution and seek to model the actions in the dataset with a Gaussian, for other settings we simply have $f_\ell$ predict the actions directly (i.e. predicting a deterministic estimate of the actions rather than a distribution). Note that simply training $f_\ell$ to predict the actions directly, rather than setting $f_\ell$ to a generative model that seeks to model the entire action distribution, is consistent with Proposition 4—we aim to estimate a *sample* from the posterior distribution, for which it suffices to just fit a deterministic quantity, rather than fitting the entire *distribution* as generative modeling typically aims to do. Furthermore, fitting a simple predictor on the actions directly usually requires fewer training iterations than fitting, for example, a diffusion model to the entire distribution, so this also reduces the computation required to fit the ensemble.

We found in practice that using bootstrap sampling to generate the datasets $\mathfrak{D}_\ell$ in Algorithm 2 performs better than

adding noise to the dataset as Proposition 4 suggests. We use both trajectory-level or state-action-level bootstrapping. For trajectory-level bootstrapping we generate $\mathfrak{D}_\ell$ as in Algorithm 3. For state-action-level bootstrapping we generate $\mathfrak{D}_\ell$

---

**Algorithm 3** Trajectory-Level Bootstrap Sampling

---

1: **input:** demonstration dataset $\mathfrak{D}$
2: $\mathfrak{D}_\ell \leftarrow \emptyset$
3: **for** $t = 1, 2, \ldots$, number of trajectories in $\mathfrak{D}$ **do**
4:     Sample trajectory $\tau \sim \mathrm{unif}(\mathfrak{D})$
5:     $\mathfrak{D}_\ell \leftarrow \mathfrak{D}_\ell \cup \{\tau\}$
6: **return** $\mathfrak{D}_\ell$

---

as in Algorithm 4. In all experiments we parameterize our final policy with a diffusion model. Given this, Algorithm 1 is

---

**Algorithm 4** Trajectory-Level Bootstrap Sampling

---

1: **input:** demonstration dataset $\mathfrak{D}$
2: $\mathfrak{D}_\ell \leftarrow \emptyset$
3: **for** $t = 1, 2, \ldots, |\mathfrak{D}|$ **do**
4:     Sample state-action pair $(s, a) \sim \mathrm{unif}(\mathfrak{D})$
5:     $\mathfrak{D}_\ell \leftarrow \mathfrak{D}_\ell \cup \{(s, a)\}$
6: **return** $\mathfrak{D}_\ell$

---

trained on the standard diffusion loss. Further details on each experiment are given below.

To leave room for RL improvement (i.e. to ensure performance is not saturated by the pretrained policy) we limit the number of demos per task in the pretraining dataset, for both the `Robomimic` and `Libero` experiments (see below for the precise number of trajectories used in pretraining).

For the `Robomimic` experiments, we use an MLP-based architecture, pretrain on a single-task demonstration dataset, and rely on state-based observations. For `Libero`, we utilize a diffusion transformer architecture due to (Dasari et al., 2024) and rely on image-based observations and language task conditioning. In `Libero`, we pretrain a single $\hat{\pi}^{\mathrm{pt}}$ policy on the demonstration data from all 16 tasks (Black et al., 2024; Kim et al., 2024; Khazatsky et al., 2024), and then run RL finetuning on each individual task. On the WidowX experiments we utilize a U-Net architecture with image observations.

For DSRL and DPPO we utilize the publicly available implementations without modification. All policies are evaluated with 200 rollouts for `Robomimic` and 100 for `Libero`.

### D.1. Robomimic Experiments

We instantiate $\hat{\pi}^\theta$ with a diffusion policy that uses an MLP architecture. For $f_\ell$, we train an MLP to simply predict the action directly in $\mathfrak{D}_i$ (i.e. we do not use a diffusion model for $f_\ell$), but use the same architecture and dimensions for $f_\ell$ as the diffusion policies. We used trajectory-level bootstrapped sampling (Algorithm 3) to compute the ensemble. In all cases we pretrain on the Multi-Human `Robomimic` datasets, and in cases where we use less than the full dataset, we randomly select trajectories from the dataset to train on, using the same trajectories for each approach.

For each RL finetuning method, we sweep over the same hyperparameters for each pretrained policy method (i.e. BC, $\sigma$-BC, POSTBC), and include results for the best one. For $\sigma$-BC, we swept over values of $\sigma$ and included results for the best-performing one. For all experiments results are averaged over 5 seeds (we pretrain 5 policies for each approach and run RL finetuning on each of them once, for a total of 5 RL finetuning runs per pretraining method, finetuning method, and task). For each evaluation, we roll out the policy 200 times. For DPPO we utilize the default hyperparameters as stated in (Ren et al., 2024), and utilize DDPM sampling. For VALUEDICE, we use the officially published codebase, and the default hyperparameters provided there. We found that the IQL training on the data produced by VALUEDICE could be somewhat unstable, and so to improve stability, for `Lift`, added LayerNorm to the IQL critic. For DSRL, we utilize a -1/0 success reward, and otherwise utilize a 0/1 success reward, using Robomimic's built-in success detector to determine the reward. We provide hyperparameters for the individual experiments below.

*Table 6.* **Common DSRL hyperparameters for all experiments.**

| Hyperparameter | Value |
| --- | --- |
| Learning rate | 0.0003 |
| Batch size | 256 |
| Activation | Tanh |
| Target entropy | 0 |
| Target update rate ($\tau$) | 0.005 |
| Number of actor and critic layers | 3 |
| Number of critics | 2 |
| Number of environments | 4 |

*Table 7.* **DSRL hyperparameters for `Robomimic` experiments.**

| Hyperparameter | Lift | Can | Square |
| --- | --- | --- | --- |
| Hidden size | 2048 | 2048 | 2048 |
| Gradient steps per update | 20 (POSTBC, BC), | 20 (POSTBC, BC), | 20 (POSTBC, BC), |
| | 10 ($\sigma$-BC) | 10 ($\sigma$-BC) | 10 ($\sigma$-BC) |
| Noise critic update steps | 20 | 10 | 10 |
| Discount factor | 0.99 | 0.99 | 0.999 |
| Action magnitude | 1.5 | 1.5 | 1.5 |
| Initial steps | 24000 | 24000 | 32000 |

*Table 8.* **Hyperparameters for pretrained policies for `Robomimic` DSRL experiments.**

| Hyperparameter | Lift | Can | Square |
| --- | --- | --- | --- |
| Dataset size (number trajectories) | 5 | 10 | 30 |
| Action chunk size | 4 | 4 | 4 |
| train denoising steps | 100 | 100 | 100 |
| inference denoising steps | 8 | 8 | 8 |
| Hidden size | 512 | 1024 | 1024 |
| Hidden layers | 3 | 3 | 3 |
| Training epochs | 3000 | 3000 | 3000 |
| Ensemble size (POSTBC) | 100 | 100 | 100 |
| Ensemble training epochs (POSTBC) | 10000 | 6000 | 3000 |
| Posterior noise weight $\alpha$ (POSTBC) | 1 | 0.5 | 1 |
| Uniform noise $\sigma$ ($\sigma$-BC) | 0.1 | 0.1 | 0.05 |

*Table 9.* **Best-of-$N$ hyperparameters for `Robomimic` experiments.**

| Hyperparameter | Lift | Can | Square |
|---|---|---|---|
| Total gradient steps | 2000000 | 2000000 | 2000000 |
| IQL $\tau$ (1000 rollouts) | 0.5 (Bc, PostBc), 0.7 ($\sigma$-Bc), 0.9 (DICE) | 0.7 | 0.7 |
| IQL $\tau$ (2000 rollouts) | 0.5 (Bc), 0.7 ($\sigma$-Bc,PostBc), 0.9 (DICE) | 0.7 | 0.7 |
| Discount factor | 0.999 | 0.999 | 0.999 |

*Table 10.* **Hyperparameters for pretrained policies for `Robomimic` Best-of-$N$ experiments.**

| Hyperparameter | Lift | Can | Square |
|---|---|---|---|
| Dataset size (number trajectories) | 20 | 300 | 300 |
| Action chunk size | 1 | 1 | 1 |
| Train denoising steps | 100 | 100 | 100 |
| Hidden size | 512 | 1024 | 1024 |
| Hidden layers | 3 | 3 | 3 |
| Training epochs | 3000 | 3000 | 3000 |
| Ensemble size (PostBc) | 100 (1000 rollouts), 10 (2000 rollouts) | 10 | 10 |
| Ensemble training epochs (PostBc) | 3000 | 500 | 500 |
| Posterior noise weight $\alpha$ (PostBc) | 1 (1000 rollouts), 2 (2000 rollouts) | 1 | 1 (1000 rollouts), 2 (2000 rollouts) |
| Uniform noise $\sigma$ ($\sigma$-Bc) | 0.1 | 0.025 | 0.025 |

*Table 11.* **Hyperparameters for pretrained policies for `Robomimic` DPPO experiments.**

| Hyperparameter | Lift | Can | Square |
|---|---|---|---|
| Dataset size (number trajectories) | 5 | 10 | 30 |
| Action chunk size | 4 | 4 | 4 |
| train denoising steps | 100 | 100 | 100 |
| Hidden size | 512 | 1024 | 1024 |
| Hidden layers | 3 | 3 | 3 |
| Training epochs | 3000 | 3000 | 3000 |
| Ensemble size (PostBc) | 100 | 100 | 10 |
| Ensemble training epochs (PostBc) | 3000 | 6000 | 3000 |
| Posterior noise weight $\alpha$ (PostBc) | 0.5 | 0.25 | 1 |
| Uniform noise $\sigma$ ($\sigma$-Bc) | 0.1 | 0.05 | 0.05 |

### D.2. Libero Experiments

For Libero, we utilize the transformer architecture from (Dasari et al., 2024) for $\widehat{\pi}^\theta$. For POSTBC we use state-action bootstrap sampling (Algorithm 3) to generate $\mathfrak{D}_\ell$. For $f_\ell$, we utilize the same ResNet and tokenizer as $\widehat{\pi}^\theta$, but simply utilize a 3-layer MLP head on top of it—trained to predict the actions directly—rather than a full diffusion transformer. For the Best-of-$N$ experiments, POSTBC utilizes a diagonal posterior covariance estimate (that is, instead of computing the full covariance matrix as prescribed by Algorithm 2, we compute the covariance dimension-wise, and construct a diagonal covariance matrix from this), while for the DSRL runs it is trained with the full matrix posterior covariance estimate. We

train on `Libero 90` data from the first 3 scenes of `Libero 90`—KITCHEN SCENE 1, KITCHEN SCENE 2, and KITCHEN SCENE 3—and use 25 trajectories from each task in each scene. For task conditioning, we conditioning $\hat{\pi}^{\theta}$ on the BERT language embedding (Devlin et al., 2019) of the corresponding text given for that task in the Libero dataset.

For each RL finetuning method, we sweep over the same hyperparameters for each pretrained policy method (i.e. BC, $\sigma$-BC, POSTBC), and include results for the best one. We utilize the DSRL-SAC variant of DSRL from (Wagenmaker et al., 2025a). For $\sigma$-BC, we swept over values of $\sigma$ and included results for the best-performing one. The DSRL experiments are averaged over 3 different pretraining runs per method, and one DSRL run per pretrained run. The Best-of-$N$ experiments are averaged over 2 different pretraining runs per method. For each evaluation, we roll out the policy 100 times. In all cases, we utilize a -1/0 success reward, using Libero's built-in success detector to determine the reward.

We provide hyperparameters for the individual experiments below.

*Table 12.* **DSRL hyperparameters for all Libero experiments.**

| Hyperparameter | Value |
| --- | --- |
| Learning rate | 0.0003 |
| Batch size | 256 |
| Activation | Tanh |
| Target entropy | 0 |
| Target update rate ($\tau$) | 0.005 |
| Number of actor and critic layers | 3 |
| Layer size | 1024 |
| Number of critics | 2 |
| Number of environments | 1 |
| Gradient steps per update | 20 |
| Discount factor | 0.99 |
| Action magnitude | 1.5 |
| Initial episode rollouts | 20 |

*Table 13.* **Best-of-$N$ hyperparameters for all Libero experiments.**

| Hyperparameter | Value |
| --- | --- |
| IQL learning rate | 0.0003 |
| IQL batch size | 256 |
| IQL $\beta$ | 3 |
| Activation | Tanh |
| Target update rate | 0.005 |
| $Q$ and $V$ number of layers | 2 |
| $Q$ and $V$ layer size | 256 |
| Number of critics | 2 |
| $N$ (Best-of-$N$ samples) | 32 |
| IQL gradient steps | 50000 |
| IQL $\tau$ | 0.9 |
| Discount factor | 0.99 |

*Table 14.* **Hyperparameters for DiT diffusion policy in Libero experiments.**

| Hyperparameter | Value |
| --- | --- |
| Batch size | 150 |
| Learning rate | 0.0003 |
| Training steps | 50000 |
| LR scheduler | cosine |
| Warmup steps | 2000 |
| Action chunk size | 4 |
| Train denoising steps | 100 |
| Inference denoising steps | 8 |
| Image encoder | ResNet-18 |
| Hidden size | 256 |
| Number of Heads | 8 |
| Number of Layers | 4 |
| Feedforward dimension | 512 |
| Token dimension | 256 |
| Ensemble size (POSTBC) | 5 |
| Ensemble training steps (POSTBC) | 25000 |
| Ensemble layer size | 512 |
| Ensemble number of layers | 3 |
| Posterior noise weight (POSTBC) | 2 (DSRL run), 4 (Best-of-$N$ run) |
| Uniform noice $\sigma$ ($\sigma$-BC) | 0.05 |

## D.3. WidowX Experiments

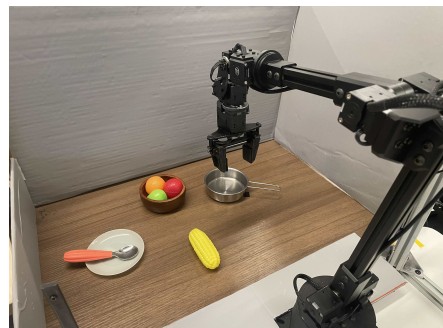

*Figure 8.* Setup for WidowX "`Put corn in pot`" task.

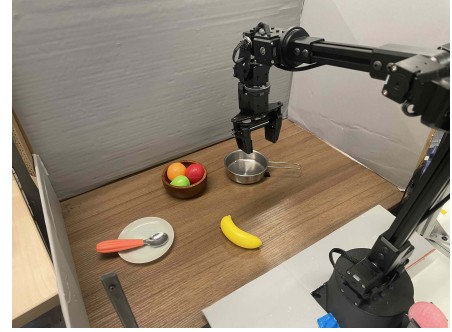

*Figure 9.* Setup for WidowX "`Pick up banana`" task.

For the WidowX experiment, we collect 10 demonstrations on the "`Put corn in pot`" task. For the diffusion policy, we utilize a U-Net architecture with ResNet image encoder. For the POSTBC ensemble predictors, we utilize a ResNet image encoder with MLP regression head, trained to directly predict the action in the dataset. For both BC and POSTBC, we pretrain the policy on the 10 demonstrations, then roll out the pretrained policy 100 times on each task, manually resetting the scene each time and classifying each trajectory as success or failure. We utilize a 0/1 reward (every step is given a reward of 0 unless it succeeds, when it is given a reward of 1). We then train an IQL $Q$-function on the rollout data and, at test time, roll out the pretrained policy, sampling $N$ actions at each step, and choosing the action with the maximum $Q$-value. For IQL, we utilize an MLP-based architecture, and to process the images, we utilize image features from a ResNet encoder pretrained on the Bridge v2 dataset (Walke et al., 2023). For both BC and POSTBC, we try different values of $N$ and different number of IQL training steps, and report the results for the best-performing values for each approach. All hyperparameters for the diffusion policy are given in Table 15, and for IQL in Table 16.

*Table 15.* **Hyperparameters for pretrained policies for WidowX experiments.**

| Hyperparameter | Both WidowX tasks |
|---|---|
| Action chunk size | 1 |
| Train denoising steps | 100 |
| Inference denoising steps | 16 |
| Image encoder | ResNet-18 |
| U-Net channel size | $[256, 512, 1024]$ |
| U-Net kernel size | 5 |
| Training epochs | 800 |
| Ensemble predictor hidden size | 512 |
| Ensemble predictor hidden layers | 3 |
| Ensemble size (POSTBC) | 10 |
| Ensemble training epochs (POSTBC) | 300 |
| Posterior noise weight $\alpha$ (POSTBC) | 1 |

*Table 16.* **Best-of-$N$ hyperparameters for WidowX experiments.**

| Hyperparameter | Put corn in pot | Pick up banana |
|---|---|---|
| IQL learning rate | 0.0003 | 0.0003 |
| IQL batch size | 256 | 256 |
| IQL $\beta$ | 3 | 3 |
| Activation | Tanh | Tanh |
| Target update rate | 0.005 | 0.005 |
| $Q$ and $V$ number of layers | 2 | 2 |
| $Q$ and $V$ layer size | 256 | 256 |
| Number of critics | 2 | 2 |
| $N$ (Best-of-$N$ samples) | 4 | 16 |
| IQL gradient steps | 400000 (BC), 700000 (POSTBC) | 100000 |
| IQL $\tau$ | 0.7 | 0.7 |
| Discount factor | 0.97 | 0.97 |

### D.4. Additional Ablations

For all ablation experiments, other than the hyperparameter we vary, we utilize the hyperparameters given in Section D.1. In Figure 12 we provide an additional ablation on the dataset size for Robomimic Square, and in Figure 13 provide additional qualitative results on Libero.

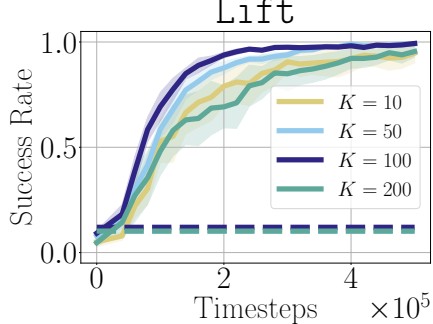

*Figure 10.* Sensitivity of POSTBC with DSRL finetuning to ensemble size. Dashed lines denote pretrained policy performance.

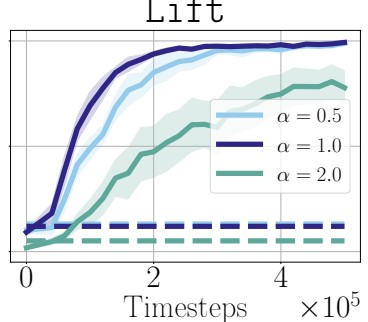

*Figure 11.* Sensitivity of POSTBC with DSRL finetuning to posterior weight. Dashed lines denote pretrained policy performance.

We consider the sensitivity of POSTBC to two key hyperparameters: the size of the ensemble ($K$) and the weight of the posterior variance ($\alpha$ in Algorithm 1). We illustrate the performance of POSTBC on `Robomimic Lift` varying these parameters in Figures 10 and 11. In Figure 10 we see that POSTBC performs best with a moderately sized ensemble ($K = 100$), but is not particularly sensitive to ensemble size as long as it is not too small or too large. In Figure 11 we see that setting $\alpha$ too large can hurt the performance of POSTBC, but that otherwise the performance of POSTBC is relatively stable with respect to $\alpha$. We note as well that setting $\alpha$ too large not only hurts the performance of the RL finetuning, but also causes the pretrained performance to drop. This is to be expected—even with the carefully tuned noise POSTBC adds to the policy in pretraining, if the weight of this noise is too large the perturbations it induces will cause performance to drop below that of the BC policy. In general, throughout our experiments, we found that $\alpha = 1$ typically performs well.

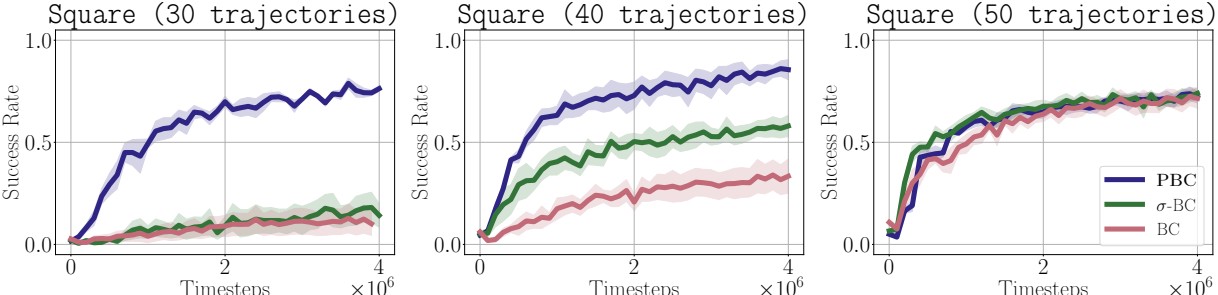

*Figure 12.* Comparison of DSRL finetuning performance combined with different BC pretraining approaches on `Robomimic Square`, varying the number of trajectories in the dataset the policies are pretrained on. As can be seen, the finetuning performance of policies pretrained with POSTBC is largely unaffected by the size of the pretraining dataset, while BC and $\sigma$-BC are both very sensitive to dataset size. For large enough datasets (50 trajectories), BC and $\sigma$-BC perform as well as POSTBC. This is to be expected—if we train on enough data, our uncertainty will be low, so POSTBC will essentially reduce to BC. These results illustrate that POSTBC gracefully interpolates between settings where BC overfits to small amounts of data, hurting its finetuning performance, and settings where BC is sufficient for effective finetuning.

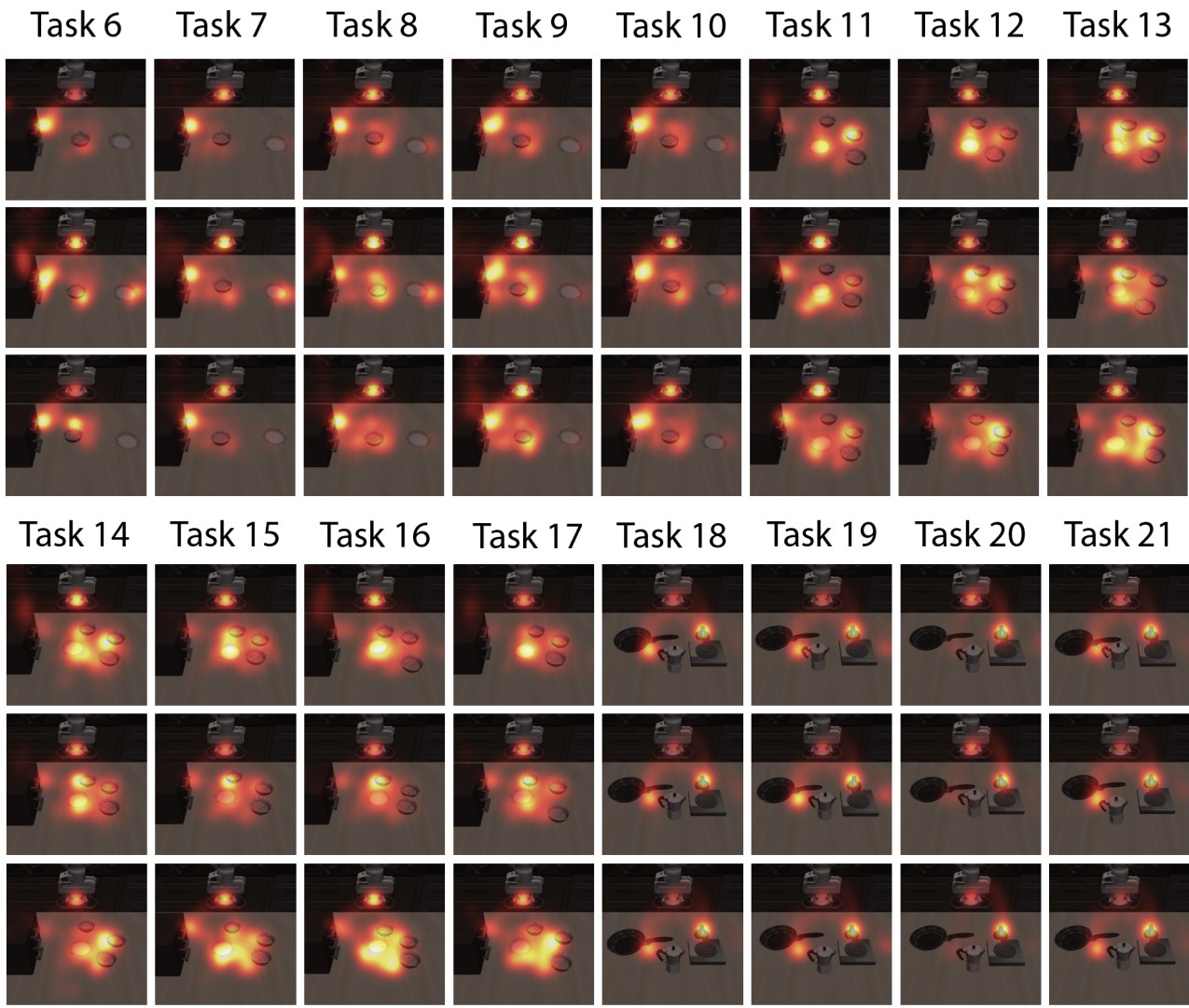

*Figure 13.* Additional density heatmaps of pretrained policies on tasks 6-21 from `Libero 90`. See Table 17 for task commands.

*Table 17.* **Task descriptions for Libero tasks in `Kitchen Scene 1-3`.**

| Task ID | Task description |
|---------|------------------|
| Task 6  | Open the bottom drawer of the cabinet |
| Task 7  | Open the top drawer of the cabinet |
| Task 8  | Open the top drawer of the cabinet and put the bowl in it |
| Task 9  | Put the black bowl on the plate |
| Task 10 | Put the black bowl on top of the cabinet |
| Task 11 | Open the top drawer of the cabinet |
| Task 12 | Put the black bowl at the back on the plate |
| Task 13 | Put the black bowl at the front on the plate |
| Task 14 | Put the middle black bowl on the plate |
| Task 15 | Put the middle black bowl on top of the cabinet |
| Task 16 | Stack the black bowl at the front on the black bowl in the middle |
| Task 17 | Stack the middle black bowl on the back black bowl |
| Task 18 | Put the frying pan on the stove |
| Task 19 | Put the moka pot on the stove |
| Task 20 | Turn on the stove |
| Task 21 | Turn on the stove and put the frying pan on it |

