# OpenReview forum: "Posterior Behavioral Cloning: Pretraining BC Policies for Efficient RL Finetuning"
_ICML.cc/2026/Conference — ICML 2026 spotlight_

### Official Review · Reviewer_qgo3 · 2026-02-20

**Soundness:** 3
**Presentation:** 4
**Significance:** 3
**Originality:** 3
**Overall Recommendation:** 4
**Confidence:** 4

**Summary:**

The paper proposes a method to train a BC policy for efficient RL finetuning.
They train a policy to model the posterior distribution of the dataset by adding noises.
Both theoretical analysis and experiments show the effectiveness of the approach.

**Compliance With Llm Reviewing Policy:**

Affirmed.

**Final Justification:**

After considering both the paper and the authors’ rebuttal, I maintain my **weak accept** recommendation. I find the paper technically solid, clearly presented, and practically appealing: the method is simple, well motivated, and supported by both theoretical analysis and empirical gains over standard BC pretraining for subsequent RL finetuning.

My main concerns were about the practical tradeoff introduced by ensemble-based pretraining, the role of ensemble uncertainty, and whether this uncertainty should also be used during online RL. The rebuttal addressed these points well by clarifying the pretraining versus finetuning time tradeoff, explaining why the ensemble variance is intended to capture epistemic rather than aleatoric uncertainty, and arguing convincingly why this signal is more appropriate for pretraining than as an intrinsic reward during RL finetuning. These responses strengthened my confidence in the paper, though they did not substantially change my overall score.

Overall, I view this as a strong and useful contribution that others are likely to build on, with some practical limitations in computational overhead, but sufficient technical merit and clarity to support acceptance.

**Key Questions For Authors:**

1. I noticed that the required ensemble size can be quite large for certain environments. Could you report the total wall-clock time to convergence for the results in Figure 2?
2. Should we also compare against using the ensemble as a signal for RL fine-tuning, for example as an intrinsic reward? I understand that time is limited, so it would be helpful if you could briefly discuss why you chose to leverage uncertainty only during the pre-training stage.
3. Could you clarify whether the ensemble is intended to capture aleatoric or epistemic uncertainty? In settings where the ensemble size is small, it is difficult to assess how reliable the ensemble-based uncertainty estimates are, so a discussion of this point would be valuable.

**Limitations:**

yes

**Strengths And Weaknesses:**

**Strengths:**
- The problem is well formulated and theoretical analysis is convincing.
- The implementation is quite straightforward and easy to implement.
- The experiments show clear improvement over pretraining BC policy in a vanilla way.

**Weaknesses:**
- Although the method is sample efficient in post training, there may exists a tradeoff as we could need large ensemble size , hyperparameter search on $\alpha$, etc.
- If the ensemble is overfitted, the $\text{conv}(s)$ could be inaccurate. Though the paper say $\alpha=1$ is typically suitable for most of the settings, the scale of $\text{conv}(s)$ is still a black box. It is hard to distinguish epistemic uncertainty and aleatoric uncertainty.

---

> ### Author Rebuttal · Authors · 2026-03-30
>
> We would like to thank the reviewer for their helpful comments and will work to incorporate their feedback in the final version of the paper. We provide detailed responses to the reviewer’s concerns below.
>
> > Although the method is sample efficient in post training..
>
> Please see Section D.4 for analysis of hyperparameter sensitivity—in general we found performance relatively stable with respect to the new hyperparameters, and did not find them any more difficult to tune than the standard hyperparameters one typically tunes in pretraining.
>
> > If the ensemble is overfitted…
>
> While in principle this is possible, in practice in all our settings (both standard robotic manipulation simulators and real-world experiments) we did not find this to be an issue. As is standard in ML, network sizes should be chosen appropriately to avoid excessive overfitting, but we did not find this to be an issue in particular for our approach. In general we found that training each member of the ensemble with networks that are approximately the same size as those that would be used for BC did not result in overfitting.
>
> > I noticed that the required ensemble size…
>
> Note first that the wall-clock time for the online RL curves in Figure 2 are approximately identical for each pretraining approach. Each pretraining approach produces a single policy in the end, so once we reach the RL finetuning phase, the training time depends only on the RL finetuning approach. Across all approaches, for Robomimic Lift, RL finetuning takes approximately 8 hours, for Can 36 hours, and for Square 50 hours.
>
> The pretraining time does differ, however, due to the cost of training the ensemble. In particular, for Robomimic Lift BC pretraining takes approximately 5 minutes while PostBC pretraining takes approximately 2.5 hours, for Can BC takes 20 minutes while PostBC takes 4 hours, and for Square, BC takes 20 minutes and PostBC takes 8 hours. While PostBC pretraining can take significantly longer than BC pretraining, we note that:
>
> - RL finetuning is significantly more time-consuming than pretraining in all cases. Thus, if PostBC significantly improves RL finetuning performance, net time will actually be smaller for PostBC. In Lift, PostBC reaches a success rate of 80% approximately 2x faster than BC during RL finetuning, so total time (pretrain + finetune) to reach 80% for PostBC is approximately 6.5 hours, while for BC it is 8 hours. Similarly for Can, total time to 80% for PostBC is 22 hours, while for BC it is 36 hours. For Square, we are not able to improve the BC policy at all, while the PostBC policy can be efficiently improved. In many cases RL finetuning time is more important than pretraining time—pretraining can be performed fully offline while RL finetuning must be performed online, which, on real robotic hardware, is much more expensive—so this reduction in total time can make the additional pretraining time a negligible cost.
>
> - Pretraining time is highly implementation and hardware dependent. In principle, ensemble members can be trained in parallel, so given a large enough GPU, if all are trained in parallel the total additional pretraining cost may be relatively minor.
>
> > Should we also compare against using the ensemble as a signal for RL fine-tuning…
>
> In our setting, the uncertainty estimated by the ensemble is over the *demonstrator’s action*, while in typical intrinsic reward-based RL the uncertainty is over the *optimal Q-function*. This is an important difference since, in our case, we do not gain any more information about the demonstrator’s behavior during the online RL finetuning phase, and therefore the ensemble variance cannot be further decreased online, while in the RL case the uncertainty of the optimal Q-function can be decreased as we observe online transitions and rewards. Given this, if we used our ensemble variance as an intrinsic reward for online RL, it would lead to *over-exploration*, as the ensemble variance would never decrease, and the agent would not learn to exploit, as is desired. Thus, the correct way to use this ensemble variance is how we have used it—to quantify the uncertainty in the initial demonstrations, and incorporate this in the pretrained policy.
>
> > Could you clarify whether the ensemble…
>
> The ensemble captures epistemic uncertainty. At a given state, the ensemble will only have high variance if we have a small number of observations from that state (or nearby states)—if we have a large number of observations from that state, then the variation in training data each ensemble member trains on will average out and they will produce similar predictions. This is true regardless of the stochasticity of the distribution (i.e. aleatoric uncertainty), and this form of ensemble variance therefore captures epistemic uncertainty. Regarding the accuracy of the uncertainty estimates, please see our response to Reviewer TAPk, Q3.

---

> > ### Author Rebuttal · Reviewer_qgo3 · 2026-04-01
> >
> > Thank u for the clear responses. I have no further questions and will keep my score.

---

### Official Review · Reviewer_Z3qu · 2026-03-12

**Soundness:** 4
**Presentation:** 3
**Significance:** 4
**Originality:** 4
**Overall Recommendation:** 5
**Confidence:** 4

**Summary:**

Pretraining from demonstrations followed by reinforcement learning (RL) fine-tuning is widely used in various domains. While many existing works focus on designing more efficient fine-tuning algorithms, this paper studies a more fundamental problem: whether the pretrained policy (e.g. via behavior cloning (BC)) is a good initialization for downstream RL fine-tuning. This paper theoretically shows that while BC is near-optimal for pretrained performance, it may fail to cover demonstrator-supported actions unless the dataset is exceptionally large. Simply adding exploration noise to improve action coverage will introduce a tradeoff between action coverage and policy optimality. This paper then shows that mixing BC with the posterior distribution over demonstrator policies learned through the dataset can preserve BC’s optimality guarantees while improving action coverage. Based on this insight, the authors propose a practical algorithm implementation of mixing BC with posterior demonstrator policy, namely Posterior Behavior Cloning (POSTBC), through approximation using ensemble uncertainty and action perturbations. The authors then conducts empirical experiments on manipulation benchmarks and real robot manipulation tasks to show improved RL fine-tuning performance over BC.

**Compliance With Llm Reviewing Policy:**

Affirmed.

**Final Justification:**

After the authors clarify the failed case and the usage of Gaussian approximation, the work appears more technically solid. The method’s applicability to DPPO and other on-policy algorithms also suggests broader potential. Therefore, I recommend accepting this paper.

**Key Questions For Authors:**

1. I would be interested in seeing an analysis of failure cases where POSTBC performs worse than standard BC.

2. Additional analysis of the gap between the theoretical formulation and the practical implementation, as well as insights into why POSTBC remains effective when combined with DPPO, would further strengthen the paper.

**Limitations:**

Yes

**Strengths And Weaknesses:**

**Strengths:**
1. **Addresses a fundamental problem.**
Rather than proposing another fine-tuning algorithm, this paper studies a more foundational question: how to learn a better pretrained policy that serves as a stronger initialization for downstream RL improvement.

2. **Well-established theories.**
The paper first shows that standard BC can cover demonstrator action distributions even though achieving near-optimal pretrained performance. It then proves that POSTBC can achieve demonstrator action coverage while retaining near-optimal performance guarantees. The theoretical development is well structured, and no obvious errors were found in the proofs.

3. **Practical engineering perspective.**
Beyond theory, the authors provide a practical implementation of the proposed approach that can be applied even when the Markovian assumptions used in the theory do not strictly hold. Experiments on benchmark manipulation datasets and real-world robotic manipulation tasks demonstrate consistent improvements over standard BC across multiple RL fine-tuning algorithms.

**Weaknesses:**
1. **Failure case analysis.**
In Figure 4 (Libero Task 15), POSTBC performs worse than standard BC. A discussion or analysis of such failure cases would be helpful. E.g. is this degradation due to approximation errors in the practical implementation, dataset characteristics, or other factors?

2. **Minor typo.**
In line 854: optimal policy on $M^2$ should be $a_3$ instead of $a_2$ according to construction.

3. The practical implementation of the algorithm relies on approximation to posterior via sampling with Gaussian perturbations as well as ensembled predictors. While this works well empirically, it would be helpful to provide additional analysis of this approximation to better understand the gap between the theoretical algorithm and the practical implementation (e.g. approximation error bounds).

4. It would be helpful to study how sensitive POSTBC is to key hyperparameters, such as the ensemble size $K$, the injected noise/perturbation magnitude, and the posterior mixing weight $\alpha$.

5. The experiments show that POSTBC is still effective when combined with DPPO, even though DPPO can alter the pretrained policy. Additional discussion on why POSTBC still helps in this setting (e.g., providing better initial coverage or exploration) would strengthen the empirical insights.

**Overall,** this is a solid paper with a clear motivation, strong theoretical grounding, and convincing empirical validation. The weaknesses listed in points 3–5 are primarily suggestions for strengthening the work rather than critical flaws.

---

> ### Author Rebuttal · Authors · 2026-03-30
>
> We would like to thank the reviewer for their helpful comments and will work to incorporate their feedback in the final version of the paper. We provide detailed responses to the reviewer’s concerns below.
>
> > 1. Failure case analysis
>
> We note that Libero Task 15 is the only example across all our experiments where BC outperforms PostBC, and that on this example, none of the approaches perform effectively—while RL finetuning on PostBC is essentially unable to learn at all, RL finetuning on BC converges to a success rate only around 10%.
>
> For Libero Task 15 (the task “Put the middle black bowl on top of the cabinet”), the success rate of the base policy for all approaches is extremely low (4.3% for BC, 1.0% for $\sigma$-BC, and 1.67% for PostBC). RL finetuning is often challenging for pretrained policies that have success rate this low, which may explain the poor performance of all methods. Furthermore, the slightly higher pretrained performance of BC compared to PostBC on this task may explain why BC outperforms PostBC here.
>
> Figure 13 illustrates the visitation heatmap for each pretrained policy on this task (the first row in Figure 13 corresponds to BC, the second row to $\sigma$-BC, and the third row to PostBC). As this figure illustrates, PostBC does induce a wider action distribution around the key step in the task—picking up the correct bowl—which is the behavior we would hope for. As PostBC is generally inducing the desired behavior of expanding the action distribution at critical points, it is not clear why the pretrained performance is worse—perhaps PostBC is inducing too much diversity, leading it away from the correct behavior, and further tuning the noise weight to reduce diversity would improve the performance of PostBC in this task. We will add this failure case analysis to the final version of the paper.
>
> > 2. Minor typo
>
> Thank you for pointing out this typo. We will correct it in the final version.
>
> > 3. The practical implementation of the algorithm relies on approximation to posterior…
>
> Please see our response to Reviewer TAPk (W1 and Q1) on our use of the Gaussian approximation. While we agree that approximation error bounds would be interesting, showing this theoretically is highly non-trivial and setting-dependent, and in practice it is not clear how to obtain such error bounds since, for all experiments, the demonstrations are drawn from a human demonstrator and we do not have access to a ground truth distribution we can evaluate our approximation with respect to.
>
> We emphasize that the approach motivated by the Gaussian approximation leads to substantial improvements even when run on human demonstrations which do not satisfy our Gaussian assumption, so the approximation errors introduced by this assumption do not seem to hurt performance in practice. Furthermore, the key conceptual insight—that we should fit the posterior instead of directly imitating the demonstrations—is independent of the Gaussian assumption, and could be combined with any other approach for estimating a posterior distribution (e.g. Bayesian Neural Networks).
>
> > 4. It would be helpful to study how sensitive POSTBC is to key hyperparameters…
>
> We have, in fact, already run these ablations. Please see Appendix D.4 for the results.
>
> > 5. The experiments show that POSTBC is still effective when combined with DPPO…
>
> We would expect PostBC to improve performance even of DPPO due to its ability to adjust exploration proportional to demonstrator uncertainty. At states where we have a large number of demonstrations, the posterior variance will be small, and the PostBC policy will have converged to the (correct) demonstrator action, while at states where we do not have a large number of demonstrations, the PostBC policy will have a much higher action variance, inducing more exploration. Thus, at states where we know what to do—states where we should not explore—the PostBC policy doesn’t explore, while at state where we don’t know what to do—states where we should explore—the PostBC policy does explore. Any on-policy RL algorithm (such as DPPO) that relies on the PostBC policy can therefore take advantage of this adaptive exploration to collect observations that enable improvement. This holds even if the weights of the base policy are modified, so long as the behavior of the policy is not changed too quickly during training (which is the case for DPPO). In contrast, the BC policy will not adapt its exploration with the demonstrator’s uncertainty, and running DPPO on this policy will therefore lead to less exploration and slower improvement. We will add this discussion to the final version of the paper to make this clear.

---

> > ### Author Rebuttal · Reviewer_Z3qu · 2026-04-03
> >
> > I thank the authors for providing a failure case analysis on Libero Task 15, which clarifies that PostBC appears worse than standard BC because all methods perform poorly on this task, making the performance signal noisy. I also appreciate the deeper analysis of the Gaussian approximation and the role of PostBC in improving on-policy algorithms such as DPPO. All my questions have been addressed, and I will maintain my positive recommendation score of 5.

---

### Official Review · Reviewer_TAPk · 2026-03-12

**Soundness:** 4
**Presentation:** 4
**Significance:** 3
**Originality:** 3
**Overall Recommendation:** 5
**Confidence:** 4

**Summary:**

This paper studies a practical problem in robot learning: standard behavioral cloning (BC) is widely used for pretraining, but it may be a poor initialization for RL finetuning because it can become overconfident under limited demonstrations. To address this, the paper proposes Posterior Behavioral Cloning (POSTBC), which preserves uncertainty in low-data regions and improves downstream exploration. The method is simple, practical, and is supported by both theoretical analysis and experiments on simulation and real-robot tasks.

**Compliance With Llm Reviewing Policy:**

Affirmed.

**Final Justification:**

The rebuttal addresses my concerns reasonably well. Overall, I remain highly positive on the paper and keep my score at 5 (accept).

**Key Questions For Authors:**

Q1. What kinds of cases are not well captured by the Gaussian assumption, and how could the method be extended to handle them?

Q2. In D.4, is the improvement from 30 to 50 demonstrations mainly because the larger dataset gives better action balance, or simply because more data is enough? If the dataset is larger but still imbalanced, would the baselines also work well?

Q3. The method depends on uncertainty estimation. Is the quality of this uncertainty estimate evaluated? What factors most affect its accuracy?

**Limitations:**

Yes

**Strengths And Weaknesses:**

## Strengths
S1. Good motivation.
The paper identifies an important mismatch between standard BC pretraining and RL finetuning. This motivation is clear and meaningful.

S2. Simple method for an important problem.
The proposed method is relatively simple and practical, while addressing a useful problem. It also seems broadly applicable and has good potential for real use.

S3. Complete evaluation.
The paper includes both theoretical analysis and empirical results, which makes the overall contribution more convincing.

## Weaknesses

W1. The Gaussian assumption may be limiting.
One concern is that the practical motivation of the method is built from a Gaussian demonstrator setting, where the posterior policy and the added perturbations take a particularly clean form. However, in many real robotic tasks, the action distribution may be multimodal, highly anisotropic, or constrained by contact dynamics, which may not be well captured by a Gaussian approximation. In such cases, the estimated posterior variance may not faithfully represent the true uncertainty structure, and the resulting perturbations could be less appropriate than the theory suggests. Although the paper does not claim that all experiments strictly follow the Gaussian setting, this assumption still appears to play an important conceptual role in the method design.

W2. The benefit seems to shrink with slightly more data.
In Appendix D.4, increasing the number of demonstrations from 30 to 50 already largely closes the gap between the proposed method and the baselines. Since 50 demonstrations is not an unrealistic amount of data, this raises the question of whether simply collecting a bit more data may be enough in practice.

---

> ### Author Rebuttal · Authors · 2026-03-30
>
> We would like to thank the reviewer for their helpful comments and will work to incorporate their feedback in the final version of the paper. Regarding the Gaussian assumption, we would like to emphasize that in all our experiments the demonstrations are generated by a human and so the Gaussian assumption certainly does not hold, but our approach still performs effectively, illustrating that this assumption is not a critical limitation in practice. We provide detailed responses to the reviewer’s concerns below.
>
> ## W1 and Q1: Gaussian Assumption
>
> We would like to make several comments on the Gaussian assumption:
>
> - First and perhaps most importantly, our experimental results show that, in both realistic robotic simulators as well as real-world robotic manipulation tasks, our approach still leads to substantial gains in performance. For all our simulated and real-world experiments, the demonstrations are generated by a human demonstrator, and the Gaussian assumption certainly does not hold. Thus, in practice it does not appear that the Gaussian assumption is limiting, and the method motivated by the Gaussian assumption still performs effectively even when the demonstration data is not Gaussian.
>
> - Second, our conceptual insight—that we should fit the posterior instead of directly imitating the demonstrations—is independent of the Gaussian assumption, and could be combined with any other approach for estimating a posterior distribution (e.g. Bayesian Neural Networks). We chose to utilize the Gaussian ensemble approximation due to its simplicity and effectiveness in our setting, but combining our approach with other methods would be a straightforward extension, and would be an interesting direction for future work.
>
> - Finally, this approximation is a fairly standard approximation in the RL literature when optimistic exploration approaches are implemented (see e.g. (Osband et al., 2018)).
>
> ## W2: Dependence on data size
>
> We would like to make several comments on the dependence on data size:
>
> - First, the dependence on the data size is highly situation-dependent. For example, in the Libero experiments, we pretrain our policy on 400 demonstration trajectories total (25 per task), and find that it leads to substantial gains over standard BC. This is closer to the “generalist” regime, where policies are trained on many tasks but a small number of demonstrations per task, and our results therefore illustrate that our approach can lead to substantial gains even when there are many demonstrations, depending on how these demonstrations are distributed.
>
> - Second, in single-task settings with many demonstrations, we would expect our approach to lead to minimal gains over standard BC, but this is the desired behavior and our approach cleanly interpolates between these regimes. If we have a significant number of demonstrations the posterior variance will become very small and the posterior demonstrator policy will therefore converge to the BC policy which is what we would hope for—if we are certain of the (correct) demonstrator behavior, we should simply play it. When fewer demonstrations are present, the posterior variance will be large and the posterior demonstrator policy therefore will have much higher action coverage than the BC policy. Our approach automatically interpolates between these regimes, achieving the best of both worlds without manual tuning.
>
> ## Q2
>
> It is not entirely clear in this particular case why increasing the number of demonstrations leads to smaller gains in performance, beyond the general reasons given in response to W2. Robomimic tasks are fairly simple, short-horizon, and uni-modal, and we conjecture that after a relatively small number of demonstrations the posterior variance at all visited states has shrunk enough that the BC and PostBC policies do not differ significantly, leading to similar performance. The exact mechanism here is somewhat difficult to determine, however, and likely depends in a complex manner on the consistency and distribution of the demonstrations, and complexity of the task.
>
> ## Q3
>
> It is unclear in general how to estimate the accuracy of the uncertainty estimates, so we did not explicitly investigate this. In particular, in all our experiments the demonstrations were generated by a human demonstrator, so we do not have access to the “ground truth” distribution, and therefore cannot determine how close our estimate is to the ground truth, and whether the distance is captured by the uncertainty estimate. We do note, however, that the ensemble-based uncertainty estimation approach we utilize is a common approach in the RL literature to estimate uncertainty.

---

> > ### Author Rebuttal · Reviewer_TAPk · 2026-03-31
> >
> > The rebuttal addresses my concerns reasonably well. The authors clarify that the Gaussian assumption is mainly a motivation rather than a strict requirement, and that the method still works well on human-generated data. They also explain that the reduced gain with more demonstrations is task-dependent and is partly expected, since the method should naturally approach standard BC when uncertainty becomes small. While the uncertainty estimate itself is not directly evaluated, the limitation is acknowledged.
> >
> > Overall, I remain highly positive on the paper and keep my score at 5 (accept).

---

### Official Review · Reviewer_wNTW · 2026-03-14

**Soundness:** 3
**Presentation:** 2
**Significance:** 3
**Originality:** 3
**Overall Recommendation:** 5
**Confidence:** 3

**Summary:**

This paper contributes a policy pretraining scheme that formulates imitation learning as a problem of posterior estimation. A generative policy fits a posterior distribution that represents the behaviour policy. The base policy relies on uncertainty to trade-off exploitation of the behaviour policy and exploration. The authors validate the approach on a suite of manipulation tasks that involve picking and placing objects.

**Compliance With Llm Reviewing Policy:**

Affirmed.

**Final Justification:**

The authors addressed all concerns demonstrating the significance of their contribution. I had misunderstood some aspects of the paper (e.g., sample bound) due to unconventional notation. Some points regarding presentation (notation, nomenclature, etc.) might be addressed in future revisions. In any case, this paper brings a fresh, rigorous perspective to imitation learning.

**Key Questions For Authors:**

Questions
1. How did the authors select the 16 tasks from Libero? What are the tasks at a categorical level? Why are these an appropriate choice (what do they add to the evaluation)?
2. Can this formulation be used in more general imitation learning approaches - for example, BC with state-only trajectories or inverse RL?

**Strengths And Weaknesses:**

Strengths
1. The pretraining strategy proposed by the authors is intuitive. There do not appear to be
other methods that successfully use the author's formulation.
2. The authors state the problem and describe the method rigorously.

Suggested improvements
1. While intuitive, the posterior estimation suggested by the authors appears to be Bayesian optimisation formulation of very related OfflineRL similar works [1-2]. While it is understood that training in this setting is reward-free, it is not clear that posterior estimation will scale well with task complexity (i.e., will require too many demonstration samples).
2. Related to (1), it would be useful to express sample bound in the main paper. If there is one, it was not immediately obvious.
3. The rigor in this paper is a plus, however, the authors should consider simplifying and apply more standard notation in many parts of the main paper to prioritise salience. As an example, in the preliminaries section, the "demonstration" policy is often referred to as the "behaviour" or "expert" policy (e.g., $\pi^*$) and the other policy is the "base" policy (e.g., $\pi$). Unless $\beta$ is a parameter, it might be better to remove it. Similarly, the hat on the base policy could be omitted. The transition probabilities are conventionally stated as fixed - it was unclear whether there is an assumption that the dynamics are changing at every time step. There are many other such simplifications of notations that do not add to the statements.
5. The paper appears unfinished. There is no discussion or conclusion section.

[1] Efficient Diffusion Policies for Offline Reinforcement Learning, Kang et al., 2023 NIPS

[2] Diffusion Policies as an Expressive Policy Class for Offline Reinforcement Learning, Wang et al., 2023 ICLR

---

> ### Author Rebuttal · Authors · 2026-03-30
>
> We would like to thank the reviewer for their helpful comments and will work to incorporate their feedback in the final version of the paper. Regarding the similarity to offline RL works, we would like to highlight that our approach operates in a significantly different setting than offline RL (the RL setting where rewards are present vs the imitation learning setting with no reward), and therefore these works do not apply in our setting. Furthermore, our techniques, which rely on uncertainty estimation, differ substantially from the approaches proposed in these offline RL works. We provide more detailed responses to the reviewer’s concerns in the following.
>
> > 1. While intuitive, the posterior estimation suggested by the authors…
>
> We want to highlight two key differences with these works. First, as the reviewer notes, these works operate in the offline RL setting, where the data could be very suboptimal but rewards are given, while we operate in the imitation learning setting, where the data comes from a demonstrator that we want to imitate but no rewards are given. The difference in feedback type in these settings necessitates entirely different approaches—RL vs imitation learning—and our approach is therefore fundamentally different than offline RL methods, and offline RL methods do not directly apply in our setting.
>
> Second, our approach operates by expanding the action distribution of the pretrained policy proportional to how uncertain we are of the demonstrator’s behavior at a given state. In contrast, the offline RL works listed here do not explicitly account for uncertainty, and instead seek to find a policy that maximizes reward subject to constraints that ensure the learned behavior is in-distribution for the offline data. These approaches are therefore much more analogous to standard BC—which also does not explicitly take into account the uncertainty—while our approach introduces a fundamental innovation by incorporating uncertainty.
>
> > Q1. How did the authors select the 16 tasks from Libero?
>
> The Libero benchmark is quite large, containing 130 tasks total, and it is common practice to only run on a subset of these tasks due to the computational cost of running on all tasks (see e.g. [1, 2, 3]). In our case, we simply ran on all tasks included in the first three scenes of Libero 90 (Kitchen Scene 1-3)—these tasks were chosen for no other reason than their ordering in the Libero benchmark, and we would expect our method to perform just as well on other tasks.
>
> The three bottom images in Figure 5 illustrate the scenes these tasks are from, and Table 17 gives the task descriptions. Libero is a widely used benchmark for robotic manipulation and includes standard manipulation tasks, so we believe the Libero results show that our approach is effective in robotic manipulation regimes. In addition, unlike for Robomimic, Libero enables multi-task training, so by running on Libero we were able to illustrate that our approach scales to multitask pretraining settings.
>
> > 2. it would be useful to express sample bound in the main paper
>
> Would the reviewer be able to clarify what exactly they are looking for here? Theorem 1 expresses a sample complexity bound with respect to the demonstration data—is there something more the reviewer is looking for?
>
> > 3. the authors should consider simplifying and apply more standard notation…
>
> Thank you for this suggestion—we will work to simplify the notation in the paper for the final version.
>
> > 4. There is no discussion or conclusion section
>
> We will add a conclusion for the final version of the paper.
>
> > Q2. Can this formulation be used in more general imitation learning approaches…
>
> This is an interesting suggestion and could be an exciting direction for future work. We believe there is a direct extension of our approach to inverse RL settings—the uncertainty over demonstrator behavior we consider could directly induce uncertainty over reward function, and this could then be used to train, for example, an ensemble of policies that maximize rewards spanning this uncertainty set. A similar idea may also apply to BC with state-only trajectories, but this would require greater modification from our approach. We estimate uncertainty of the demonstrator action, but given a state-only trajectory with no actions this notion of action uncertainty does not apply. However, similar notions of uncertainty may apply in this setting, and investigating this would be an interesting future direction.
>
> [1] Pertsch, Karl, et al. "Fast: Efficient action tokenization for vision-language-action models." arXiv preprint arXiv:2501.09747 (2025).
>
> [2] Kim, Moo Jin, Chelsea Finn, and Percy Liang. "Fine-tuning vision-language-action models: Optimizing speed and success." arXiv preprint arXiv:2502.19645 (2025).
>
> [3] Lu, Guanxing, et al. "Vla-rl: Towards masterful and general robotic manipulation with scalable reinforcement learning." arXiv preprint arXiv:2505.18719 (2025).

---

> > ### Author Rebuttal · Reviewer_wNTW · 2026-04-04
> >
> > Thank you for clarifying the differences between this approach and offline RL. These arguments make sense. However, the first suggestion raises a concern about the challenges of posterior estimation for more complex tasks (e.g., beyond pick-and-place). This point remains unaddressed.
> >
> > Otherwise, all other points have been addressed. I misunderstood the sample bound, the selection of evaluation tasks is more clear, and the authors have promised to simplify the notation and write a conclusion. Finally, the response to Q2 is much appreciated.

---

> > > ### Author Response · Authors · 2026-04-04
> > >
> > > We would like to thank the reviewer for considering our rebuttal, and provide answers to their follow-up questions below.
> > >
> > > > Thank you for clarifying the differences between this approach and offline RL. These arguments make sense. However, the first suggestion raises a concern about the challenges of posterior estimation for more complex tasks (e.g., beyond pick-and-place). This point remains unaddressed.
> > >
> > > Posterior estimation can indeed be challenging, but our experimental results demonstrate that the ensemble-based approximation to the posterior we propose scales effectively to a range of standard robotic manipulation tasks. We would like to highlight several points on this:
> > >
> > > - First and perhaps most importantly, our experimental results show that this is not a major shortcoming in standard robotic manipulation settings, both simulated and real-world, and that the ensemble-based approximation to the posterior introduced in Section 5 scales effectively to these domains. In particular, Robomimic and Libero are two of the most commonly used simulated benchmarks for robotic manipulation, and the real-world tasks we consider are similar in complexity to the tasks commonly included in standard robotic manipulation datasets [1]. Furthermore, the tasks we consider encompass skills beyond pick-and-place such as turning on a stove and opening drawers, and our experimental results encompass both single-task and multi-task pretraining settings. Thus, our experimental results demonstrate that our posterior approximation approach leads to significant performance improvements across standard manipulation settings. Given this, we do not anticipate any intrinsic difficulties scaling it to more complex robotic manipulation settings—indeed, we see no fundamental reason why our approach would perform any worse on such tasks than the ones we consider.
> > >
> > > - Second, we want to emphasize that the tasks we considered were chosen only because these are the standard manipulation tasks typically considered in the robot learning community. While extending learning-based approaches for robotic manipulation to more challenging tasks is certainly of interest within the community, at present there do not exist widely-used benchmarks or standardized open-source datasets that encompass such tasks. While we are excited by the ongoing work in the robot learning community to develop benchmarks encompassing more complex tasks, given the lack of such benchmarks currently, we felt that developing such benchmark tasks to validate our approach to be out of scope for this work.
> > >
> > > - Finally, we want to highlight that there are a variety of other approaches to posterior estimation which could also be used to instantiate our key insight—that instead of simply running BC, we should estimate the demonstrator’s posterior. For example, there is a rich literature on Bayesian Neural Networks which would provide another means of estimating the posterior, and has been shown to scale to a variety of different settings. We did not consider these approaches in our submission as we found the ensemble-based approximation both very effective and easy to implement. If a setting is encountered where this approach does not perform effectively, however (though we emphasize we did not encounter such a setting), these other posterior estimation approaches would be promising approaches to try.
> > >
> > > Due to ICML’s discussion phase policy, we will not be able to respond to any further questions the reviewer has, but hope that this addresses the reviewer’s concerns.
> > >
> > > [1] Walke, Homer Rich, et al. "Bridgedata v2: A dataset for robot learning at scale." Conference on Robot Learning. PMLR, 2023.

---

### Decision · Program_Chairs · 2026-04-30

**Decision:**

Accept (spotlight)

**Comment:**

This paper tackles the problem of how best to pretrain a policy using behavoir cloning in order to provide effective RL finetuning on specific tasks. The authors argue convincingly that standard BC pretraining fits the demonstration actions too narrowly and in order to provide a wider range of actions for fine tuning propose BC pretraining as learning the posterior distribution over demonstrator actions given the observed demos.   Reviewers generally found the method to be novel and well-motivated, the presentation to be rigorous, and evaluation strong.  There were some concerns expressed regarding whether the method could scale, whether the Gaussian assumption was too limiting, and if there could be more discussion of failure modes.  The authors largely addressed these concerns in the rebuttals (and are encouraged to modify the manuscript as appropriate) leading to consensus in favor of acceptance.